# A hybrid algorithm for ship clutter identification in pulse compression polarimetric radar observations

Shuai Zhang[1,2,3], Haoran Li[4,5], Dmitri Moisseev[5,6], Leskinen Matti[5]

[1]Meteorological Observation Center, China Meteorological Administration, Beijing, China
5 [2]State Key Laboratory of Environment Characteristics and Effects for Near-space, Beijing, China
[3]Engineering Technology Research Center for Meteorological Observation, China Meteorological Administration, Beijing, China
[4]State Key Laboratory of Severe Weather, Chinese Academy of Meteorological Sciences, Beijing, China
[5]Institute for Atmospheric and Earth System Research/Physics, Faculty of Science, University of Helsinki, Helsinki, Finland
10 [6]Finnish Meteorological Institute, Helsinki, Finland

*Correspondence to*: Haoran Li (lihr@cma.gov.cn) and Dmitri Moisseev (dmitri.moisseev@helsinki.fi)

**Abstract.** With the rapid development of active-phased arrays and solid-state transmitters, pulse compression technology has become increasingly important. Currently, pulse compression waveforms with peak sidelobe levels better than -50 dB have been developed, enabling the broader application of pulse compression technology in weather radar systems. However, 15 existing sidelobe suppression levels are still insufficient to ensure that radar data quality is unaffected by range sidelobes for ship clutter, which have a high echo intensity and cannot be removed by conventional quality control methods. In this study, we introduce a Hybrid Ship Clutter Identification (HSCI) algorithm to address this issue in pulse compression polarimetric radar observations. The HSCI algorithm comprises two parts: mainlobe and sidelobe identification (including the range and antenna sidelobes). Mainlobe identification uses a random forest model that integrates multiple features to identify the 20 mainlobe of ship clutter. Sidelobe identification uses a series of heuristic criteria derived from the statistical characteristics of ship clutter to distinguish them from precipitation echoes. The analysis results of two typical cases indicate that after implementing the HSCI algorithm, the impact of ship clutter on radar data is visually imperceptible. The statistical results show that the HSCI algorithm achieves a promising performance in ship clutter mainlobe identification on a test dataset comprising 400 ship clutter gates and 2,500 range gates of precipitation echoes, with precision, recall, and F1-score all 25 exceeding 97%. Application of this algorithm to the University of Helsinki C-band dual-polarization Doppler weather radar data successfully reproduced ship tracks in the Gulf of Finland.

## 1 Introduction

As a sophisticated observation instrument, weather radar has significantly advanced research in disaster weather warning (Sandmæl et al., 2023; Chen et al., 2024), precipitation microphysics (Ho et al., 2023; Li et al., 2024), and quantitative 30 precipitation estimation (Li et al., 2023; Hanft et al., 2023). Evolving demands for meteorological applications continue to drive improvements in radar performance and data quality, thereby stimulating the development of innovative radar

technologies. Pulse compression, which modulates radar signals to increase bandwidth and thus achieve a better range resolution, is a typical example (Cook and Bernfeld, 1967; Rihaczek, 1969). However, the pulse compression technique was initially not widely adopted in the meteorological field because its associated range of sidelobes could obscure weak targets near strong ones.

The recent growing popularity of solid-state transmitters and phased array radars (Weber et al., 2021; Palmer et al., 2022; Kollias et al., 2022), has seen rapid advances in pulse compression technology. Bharadwaj and Chandrasekar (2012) proposed a combination of a continuous nonlinear frequency modulation (NLFM) waveform with a minimum integrated sidelobe level filter, and its performance was validated for reflectivity steps up to 40 dB through simulation experiments. Kurdzo et al. (2014) introduced an NLFM waveform designed using a genetic algorithm. Specifically, this approach optimized the frequency function represented by a Bezier curve to generate an NLFM waveform with low sidelobes and high range resolution. Compared with traditional windowed methods, this technique offered a sensitivity gain of approximately 3 dB. Torres et al. (2017) also used a genetic algorithm to design an NLFM waveform tailored to operational requirements, focusing on minimizing the transmission bandwidth as the primary optimization goal. Other optimization techniques used in this field include simulated annealing (Pang et al., 2015) and quadratic optimization (Argenti and Facheris, 2020). Owing to these advanced technologies, range sidelobes have been effectively suppressed, enabling the broader adoption of pulse compression technology in weather radar systems.

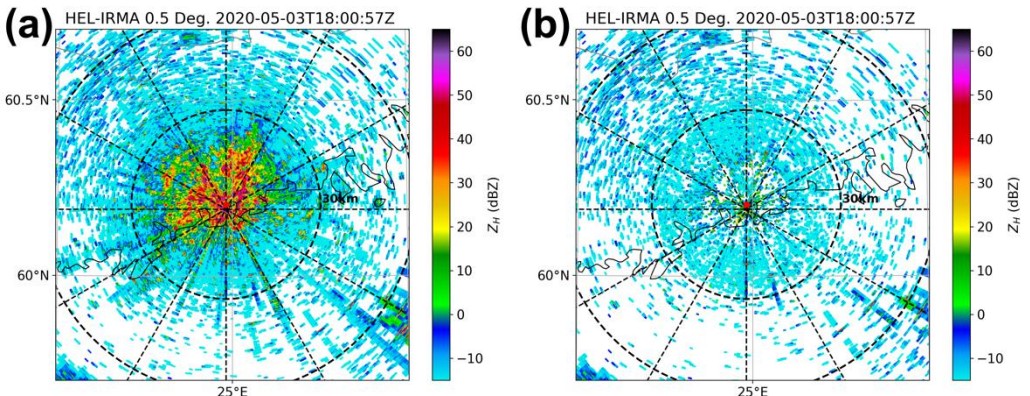

**Figure 1: 0.5° elevation of Kumpula radar using linear frequency-modulation (LFM) waveform at 1800 UTC 3 May 2020. The Gaussian model adaptive processing (GMAP) algorithm built into the RVP900 signal processor was adopted to achieve this performance. (a) Raw reflectivity; (b) Reflectivity after ground clutter filtering.**

Like conventional short-pulse radar, pulse compression radar data can also be contaminated by non-meteorological echoes. Therefore, quality control is crucial for the effective use of radar data. Ground clutter, a common type of non-meteorological echo, is caused by scattering from stationary targets, such as buildings or mountains (Billingsley, 2002). Typically, ground clutter exhibits a near-zero Doppler velocity and narrow Doppler spectrum width (Hubbert et al., 2009a). Numerous ground

clutter identification and filtering algorithms have been developed based on this characteristic (Hubbert et al., 2009b; Torres and Warde, 2014; Golbon-Haghighi et al., 2018; Hubbert et al., 2021), and have achieved substantial success (Fig. 1).

Biological echoes, caused by scattering from airborne biological entities, such as insects and birds, represent another frequent source of non-meteorological echoes (Stepanian et al., 2016). Given that the shape, orientation, and other attributes of biological targets differ significantly from precipitation particles, simple metrics such as correlation coefficients or depolarization ratio thresholds can effectively distinguish between these two types of targets (as demonstrated in Fig. 2; Kilambi et al., 2018; Pérez Hortal and Michelson, 2023).


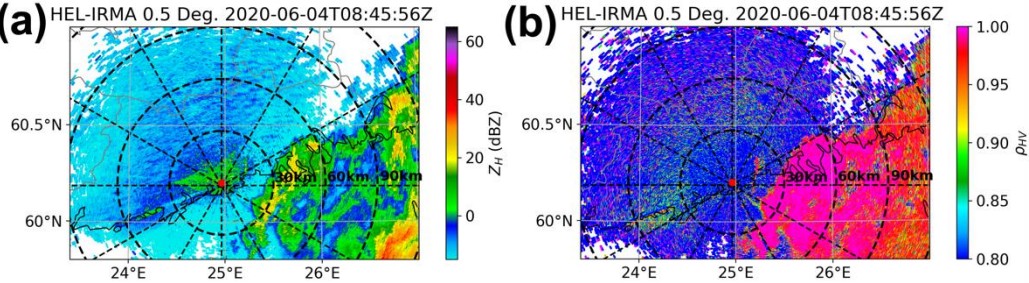

**Figure 2: 0.5° elevation of Kumpula radar using LFM waveform at 0845 UTC 4 June 2020. Precipitation echoes are concentrated within azimuthal intervals of approximately 60–180° with high correlation coefficient, while other sectors are mainly affected by biological echoes with low correlation coefficient. (a) Filtered reflectivity; (b) Correlation coefficient.**


In addition to the commonly-observed ground clutter and biological echoes, weather radars deployed along coastlines usually detect echoes scattered from ships, referred to as ship clutter, in meteorological contexts (Overeem et al., 2020). Unlike other forms of clutter, ship clutter exhibits non-zero Doppler velocities and high correlation coefficients, making it challenging for existing quality control methods to suppress it effectively (as depicted in Fig. 3). Typically, a ship spans one

or more range gates. However, in pulse compression radar systems, the impact of ship clutter is not confined to these gates but also extends radially and tangentially because of the range and antenna sidelobes. As illustrated in Fig. 3, this results in numerous cross-shaped patterns that can extend over ten kilometers on the plan position indicator (PPI), significantly compromising the quality of the radar data.

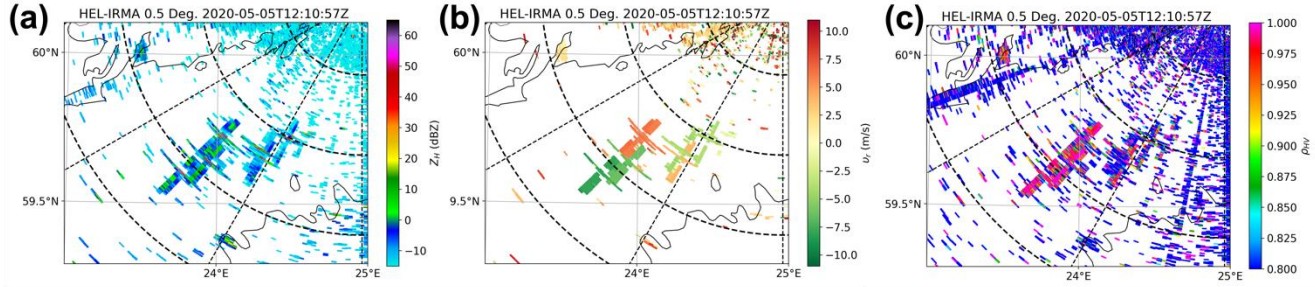

**Figure 3: 0.5° elevation of Kumpula radar using LFM waveform at 1210 UTC 5 May 2020. Several ships present cross-shaped radar variable fields, as well as non-zero Doppler velocity and high correlation coefficients. (a) Filtered reflectivity; (b) Doppler velocity; (c) Correlation coefficient.**

In this study, we propose a Hybrid Ship Clutter Identification (HSCI) algorithm to enhance the data quality and meteorological application performance of pulse compression radars. As stated above, non-meteorological echoes such as ground clutter and biological echoes can be effectively suppressed using well-established methods. This allows the identification of ship clutter in this study to be simplified as a binary classification problem, i.e., assuming that the radar observation data contains only two types of targets: ship clutter and precipitation echoes. The instruments and related datasets used in this study are described in Sect. 2. Section 3 provides an in-depth description of the HSCI algorithm, while Sect. 4 presents the algorithm performance evaluation results. Discussion and summary are presented in Sects. 5 and 6, respectively.

## 2 Instrument and data

At the Kumpula campus of the University of Helsinki, a C-band dual-polarization Doppler weather radar (hereinafter referred to as the Kumpula radar) was installed on the rooftop of the Department of Physics building (60.204°N, 24.269°E, 60 m above mean sea level). In 2019, the klystron transmitter of the Kumpula radar was upgraded to two solid-state transmitters supported by Vaisala Oyj. The radar currently serves as a prototype for evaluating the performance of solid-state transmitters and pulse compression technology. Observational data collected between May and June 2020 were used in this study.

The archived data from the Kumpula radar include the reflectivity factor at horizontal polarization ($Z_H$), Doppler velocity ($v_r$), Doppler spectrum width ($\sigma_v$), differential reflectivity ($Z_{DR}$), differential phase ($\phi_{DP}$), and co-polar correlation coefficient ($\rho_{HV}$). The scanning strategy used by the Kumpula radar diverges from that used in operational radars, such as the volume coverage pattern 21 used by the Weather Surveillance Radar-1988 Doppler (Crum and Alberty, 1993). Specifically, the Kumpula radar conducts three PPI scans at an elevation of 0.5°, using diverse transmitting waveforms: 1) unmodulated short pulse (SP), LFM, and NLFM. For the LFM and NLFM waveforms, a frequency diversity technique was applied to address the blind-zone issue (Bharadwaj and Chandrasekar, 2012). This involves transmitting an additional unmodulated

short pulse (ASP) at a slightly shifted frequency to cover the blind zones created by the modulated pulses. The detailed system characteristics of the Kumpula radar and settings for different waveforms are listed in Table 1.

Table 1: System characteristics and waveform settings of the Kumpula radar.

| | SP | LFM | NLFM | ASP |
|---|---|---|---|---|
| Pulse width (µs) | 4.5 | 90 | 90 | 1 |
| Swept bandwidth (MHz) | / | 2 | 3.8 | / |
| Pulse repetition frequency (Hz) | 600 | 800 | 800 | 800 |
| Transmitter type | Dual solid-state amplifiers | | | |
| Polarization | Dual linear | | | |
| Frequency (GHz) | 5.6 ~ 5.65 | | | |
| Peak power (kW) | 4.5 | | | |
| Maximum duty cycle | 30% | | | |
| 3-dB beam width (∘) | 1 | | | |
| Antenna diameter (m) | 4.2 | | | |
| Antenna gain (dB) | 45 | | | |
| Sample number | 40 | | | |
| Range gate spacing (m) | 150 | | | |

The Kumpula radar, situated on the north coast of the Gulf of Finland, which is a crucial waterway in Northern Europe, frequently detects ship clutter at low elevation angles. We compiled a radar dataset for ship clutter by manually identifying the distinct strong point echoes and cross-shaped signatures. When these static signatures were insufficient to confirm the presence of ship clutter, the movement of echoes across consecutive scans was used as supplementary evidence. We acknowledge that an objective approach could be more comparable to future studies, but no such methods are available. Regarding this process, a visual example can be seen in Fig. 4, from which it can be observed several typical ship clutters (marked by dashed circles with different colors) were observed by Kumpula radar at 1806 UTC 11 June 2020. Although their echo intensity and shape changed considerably over the ensuing 10 minutes (e.g., the ship clutter marked by red dashed circles), the continuous tracing proves that they came from the same target. Given that ships typically take several hours to traverse the effective field of view of the radar, we extracted only one scan per hour to maintain a high level of diversity and independence among the different instances of ship clutter. Ultimately, this method yielded a dataset comprising nearly 1600 ship clutter events across 110 scans. To evaluate the consistency of the subjectively annotated dataset, a secondary verification was conducted by an independent annotator on a representative subset. Specifically, 170 ship clutter from 10 scans were re-labeled, and differences in labeling were observed only in two relatively weak cases between the initial and secondary annotations. This result suggests that the overall annotation quality is high and that potential subjectivity has a

limited impact on the dataset. It is worth noting that the detectability of ship clutter is dependent on atmospheric conditions, and super refraction is favorable for the radar beam to detect ships. The super refraction frequency in summer is usually higher than in winter. Therefore, we are focused on analyzing summertime observations in this study.


**Figure 4: Typical ship clutter observed by Kumpula radar at (a) 1806 UTC, (b) 1811 UTC, and (c) 1816 UTC on 11 June 2020. Dashed circles with different colors indicate ship clutters, respectively.**

The precipitation dataset was manually extracted from four precipitation events that occurred on May 10, May 15, June 4, and June 5, 2020. Despite the Kumpula radar using pulse compression technology, which inherently produces range sidelobes, its peak sidelobe level was maintained below -50 dB (as shown in Sect. 3.2.1 below). Furthermore, the range sidelobes from strong precipitation echoes are typically overshadowed by the surrounding medium-intensity precipitation echoes. Consequently, for most precipitation echoes, the impact of range sidelobes was not significant. It is important to note

that both datasets—ship clutter and precipitation—were derived from observational results obtained using the LFM waveform and extracted by the same annotator who consistently adheres to the criteria mentioned above.

**3 Method**

    Figure 5 shows a flowchart outlining the HSCI algorithm, which uses a straightforward sequential structure. The process begins with the identification of the mainlobe of the ship clutter in the radar data. If the identification results indicate the

presence of ship clutter, the procedure continues with the identification of ship clutter sidelobes, followed by the removal of the entire ship clutter (i.e., both the mainlobe and sidelobes). The range gate exhibiting the highest reflectivity within the ship clutter is regarded as the mainlobe, whereas the remaining range gates are considered sidelobes.

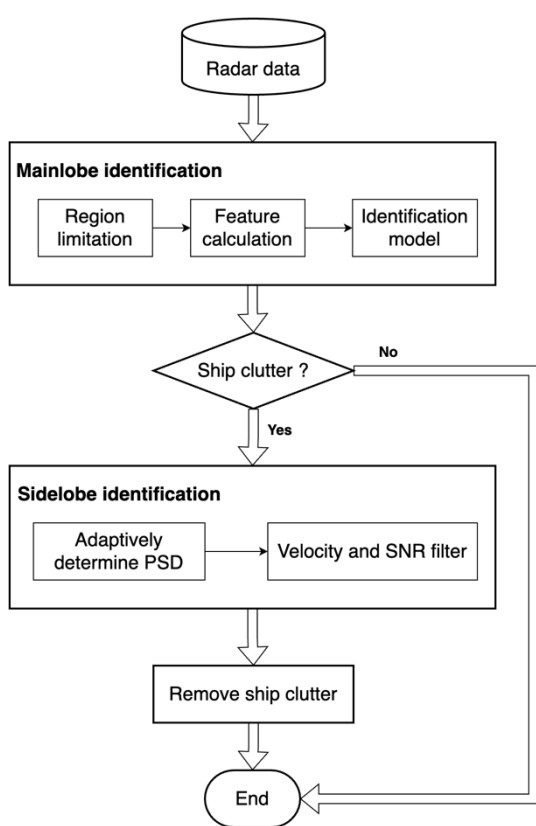

**Figure 5: Flowchart of the HSCI algorithm.**

## 3.1 Mainlobe identification

### 3.1.1 Region limitation

A single scan from weather radar typically yields hundreds of thousands of range gate observations. If the algorithm
processes each gate individually, its computational efficiency is low. Moreover, because precipitation echoes occur more
frequently and cover larger areas than ship clutter, there is a heightened risk of misidentifying these echoes as ship clutter.
Therefore, narrowing the identification area is crucial.

The HSCI algorithm incorporates three specific constraints to enhance efficiency and accuracy:

1) Identification is conducted only over sea areas, as this is naturally the most likely location for ship clutter.

2) Identification of range gates with local maximum reflectivity is based on the definition used in this study, in which the
mainlobe is the range gate with the highest reflectivity within the ship clutter.

3) A reflectivity threshold is set (defaulting to 20 dBZ) to mask weak echoes. This constraint is primarily based on the
consideration that the sidelobe of weak ship clutter are often undetectable due to the radar's sensitivity limitations. As a result,

the impact range of ship clutter is relatively small, exhibiting characteristics similar to point clutter, which can be effectively suppressed using point clutter mitigation algorithms (Zhang et al., 2004; Vaisala, 2016).

### 3.1.2 Feature calculation

Extracting features from radar variable fields that can differentiate between ship clutter and precipitation echoes is a crucial step in the implementation of machine learning algorithms. Upon analyzing these fields, six distinct features were identified: Reflectivity Difference (RD), Reflectivity Gradient Flag (RGF), $\sigma_v$, Spectrum Width Ratio (SWR), $Z_{DR}$, and Correlation Coefficient Difference (CCD).

(1) RD

The ship is a typical point target. As depicted in Fig. 3a, there was a sharp decrease in the reflectivity when the ship moved from being directly in the beam's mainlobe to being slightly off-center. This pattern was also observed in the pulse compression radar along the radial direction. Within the HSCI algorithm, RD was developed to quantify this phenomenon, which is defined as:

$$RD = max\ [Z_H(x,y) - Z_H(x-1,y), Z_H(x,y) - Z_H(x+1,y), Z_H(x,y) - Z_H(x,y-1), Z_H(x,y) - Z_H(x,y+1)]\ , \qquad (1)$$

where x and y represent the tangential and radial indices of the mainlobe in the radar variable field, respectively, and max denotes the maximum-value function.

(2) RGF

The principle of the RGF is like that of the RD, indicating that the reflectivity decreases as the distance from the mainlobe increases. RGF is defined as follows:

$$RGF = [Z_H(x-1,y) > Z_H(x-2,y)]\&[Z_H(x+1,y) > Z_H(x+2,y)]\&[Z_H(x,y-1) > Z_H(x,y-2)]\&[Z_H(x,y+1) > Z_H(x,y+2)]\ , \qquad (2)$$

where & represents the AND operation. It is important to note that, unlike RD and the other features, RGF yields a Boolean value.

(3) $\sigma_v$

As a rigid target, a ship exhibits extremely high velocity consistency. In contrast to the precipitation echo, which is formed by a multitude of precipitation particles within the sampling volume, the mainlobe of ship clutter displays a significantly lower $\sigma_v$, comparable even to ground clutter. Consequently, $\sigma_v$ was chosen as one of the features in the HSCI algorithm.

(4) SWR

When analyzing the $\sigma_v$ of ship clutter, a sudden change in the $\sigma_v$ values at the position of the mainlobe and its adjacent antenna sidelobes was observed such that $\sigma_v(x \pm 1, y) \gg \sigma_v(x, y)$. This phenomenon was observed by Feng and Fabry (2016). It was explained by the sharp change in the antenna phase pattern near the mainlobe and highlights a distinct characteristic of ship clutter.

Although $\sigma_v(x \pm 1, y) > \sigma_v(x, y)$, the difference is relatively minor when compared to the $\sigma_v$ of precipitation echoes. To better illustrate the relative relationship between $\sigma_v(x \pm 1, y)$ and $\sigma_v(x, y)$, we propose using the SWR, defined as follows:

$$SWR = \max \left[ \frac{\sigma_v(x+1,y)}{\sigma_v(x,y)}, \frac{\sigma_v(x-1,y)}{\sigma_v(x,y)} \right] . \tag{3}$$

(5) $Z_{DR}$

$Z_{DR}$, the ratio of reflectivity from horizontal to vertical polarization, can to some extent indicate the shape of precipitation
particles (Seliga and Bringi, 1976). During descent, raindrops encounter air resistance that causes large raindrops to split into smaller droplets. On the other hand, hail tumbles as it falls, leading to $Z_{DR}$ values close to 0 dB. Consequently, $Z_{DR}$ values for most precipitation echoes typically fall within the specific range of (-1 to 6 dB; Kumjian, 2013). However, in our analysis of ship clutter, we observed that $Z_{DR}$ values varied almost randomly across the entire range (-8 to 8 dB in Kumpula radar). Thus, $Z_{DR}$ was incorporated as a feature into the HSCI algorithm.

(6) CCD

As depicted in Figs. 2b and 3c, the mainlobe of ship clutter typically exhibits a high $\rho_{HV}$, like that of precipitation echoes, while that in the antenna sidelobes of ship clutter sharply decreases. This phenomenon can be attributed to the antenna pattern, where the horizontal and vertical polarization channels align well in the mainlobe but mismatch in the sidelobes. Consequently, this study introduces the CCD to quantify the disparity between the antenna mainlobe and the sidelobes of
ship clutter. The CCD is defined as follows:

$$CCD = \max \left[ \rho_{HV}(x,y) - \rho_{HV}(x-2,y), \rho_{HV}(x,y) - \rho_{HV}(x+2,y) \right] . \tag{4}$$

Figure 6 shows the normalized histograms for the six features of ship clutter and precipitation echoes using the datasets specified in Sect. 2. It is worth noting that all feature values are raw, with no normalization or scaling applied. The
overlapping areas of the probability distribution densities of the two echo types are listed in Table 2. This overlap quantitatively reflects the discriminatory ability of each feature, with smaller values indicating better differentiation capability. Although the statistical analysis revealed that RD offers the most effective discrimination among all the features, neither ship clutter nor precipitation echoes can be accurately distinguished by relying solely on a single feature. Thus, it is essential to integrate multiple features to further enhance the identification accuracy.


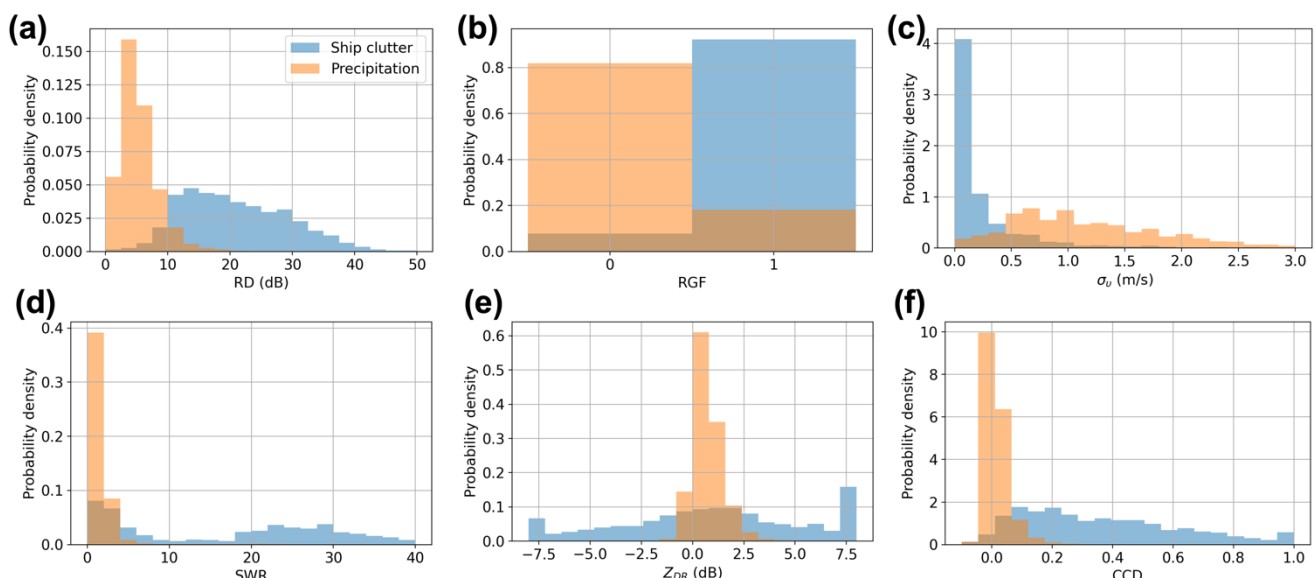

**Figure 6: Normalized histograms of six features across selected datasets. (a) RD; (b) RGF; (c) $\sigma_v$; (d) SWR; (e) $Z_{DR}$; and (f) CCD. Blue and orange denote ship clutter and precipitation echoes, respectively.**

**Table 2: Overlapping area between normalized histograms for ship clutter and precipitation echoes of six features.**

| Feature | Overlapping area |
|---|---|
| RD | 14.34% |
| RGF | 25.93% |
| $\sigma_v$ | 26.56% |
| SWR | 34.13% |
| $Z_{DR}$ | 33.35% |
| CCD | 20.19% |

### 3.1.3 Identification model

In this study, a random forest model was used to integrate multiple features to identify ship clutter. Random forest is a classic machine learning algorithm that primarily constructs multiple decision trees and combines their prediction results to 230 enhance overall prediction accuracy and stability. Owing to the advantages of random forest, such as high execution efficiency, no need to scale input features, and the ability to handle missing data, it has been widely used in the field of weather radar, including tornado identification (Sandmæl et al., 2023), precipitation forecasting (Mao and Sorteberg, 2020),

and raindrop size distribution retrievals (Conrick et al., 2020). This study does not provide an in-depth introduction to the principle of the random forest algorithm; details can be found in Ho (1998) and Breiman (2001).

Like other supervised learning methods, the development of the random forest model involves two steps: training and testing. The ship clutter and precipitation datasets mentioned in Sect. 2 were split into training and test sets at a ratio of 3:1. Although the selected precipitation dataset is extensive, approximately 10,000 range gates only remain after applying the region limitation described in Sect. 3.1.1 (7,500 for training and 2,500 for testing). In this study, the Python Scikit-learn machine learning library was used for training, testing, and subsequent prediction tasks (Pedregosa et al., 2011). The input

for the random forest model comprises the six features of the target range gate, and the output is a Boolean identification result, where 1 and 0 represent ship clutter and precipitation echoes, respectively. The hyperparameter configurations of the random forest model in the HSCI algorithm are listed in Table 3. The "GridSearchCV" method from Scikit-learn was used to determine the optimal hyperparameters (listed in the third column of Table 3) by tuning the model through iterations over the hyperparameter value ranges (shown in the second column of Table 3). Specifically, a k-fold cross-validation approach was

integrated into this process, with the default setting using 5 folds. This means that the training dataset was partitioned into five subsets, and for each hyperparameter configuration, the model was trained on four subsets and validated on the remaining one. This procedure was repeated five times (once for each subset), with the average performance across all folds serving as the evaluation metric. The hyperparameter combination that yielded the highest average performance was then selected as optimal. A detailed description of these hyperparameters can be found in Pedregosa et al., (2011).


**Table 3: List of hyperparameter values used in random forest model for HSCI algorithm.**

| Hyperparameter | Value range | Selected value |
|---|---|---|
| n_estimators | 10, 20, 50, 100, 200 | 100 |
| criterion | entropy, gini | entropy |
| max_depth | None, 5, 10 | 5 |
| min_samples_split | 1, 2, 4 | 4 |
| min_samples_leaf | 1, 2, 4 | 1 |
| min_weight_fraction_leaf | 0 | 0 |
| max_features | None, auto | auto |
| max_leaf_nodes | None | |
| min_impurity_decrease | 0 | |
| bootstrap | True | |
| oob_score | False | |
| random_state | None | |
| warm_start | False | |

### 3.2 Sidelobe identification

### 3.2.1 Adaptively determine the potential sidelobe distribution (PSD)

Although the scattered energy of the ship clutter is predominantly concentrated in the mainlobe, the extensive distribution of weaker sidelobes can significantly interfere with radar data applications. Therefore, once the mainlobe is identified, the next step involves identifying the sidelobes. Unlike mainlobe identification, which is performed gate by gate, sidelobe identification leverages mainlobe identification results to determine the PSD.

Sidelobes in ship clutter typically appear in a cross-shape but vary in size. The signal-to-noise ratio (SNR) of a sidelobe at

different positions ($SNR_{side}(\Delta x, \Delta y)$) is influenced by the SNR of the mainlobe ($SNR_{main}$), the ambiguity function, and the antenna pattern, where $\Delta x$ and $\Delta y$ represent the distance from the mainlobe in tangential and radial directions, respectively. If $SNR_{side}(\Delta x, \Delta y)$ at a range gate falls below a set SNR threshold, that gate will be masked, and no radar variables will be output. Therefore, a static PSD setting may be insufficient when $SNR_{main}$ is high, and excessive when $SNR_{main}$ is low.

To determine the sidelobe distribution settings more effectively, it is essential to make adaptive decisions for different ship

clutter events. Three main factors influence the sidelobe distribution: $SNR_{main}$, the relative power between the mainlobe and sidelobe, and the SNR threshold used in the radar variable estimation. Although the Kumpula radar does not directly output SNR values, this study proposes a method to obtain the SNR indirectly from reflectivity (details in the Appendix). Moreover, the SNR threshold is set by the user and is a known value (1.5 dB for the Kumpula radar). The relative power between the mainlobe and sidelobe can be calculated theoretically using the ambiguity function of the specified pulse compression

waveform and antenna pattern. However, discrepancies between theoretical analysis and actual observations may arise because of unforeseen factors. Consequently, this study derived the relative power between the mainlobe and sidelobe through a statistical analysis of actual data.

To capture as many sidelobe distribution characteristics as possible, only ship clutter with $SNR_{main} > 50$ dB was selected from the dataset. Because statistical results can be skewed by echoes from other sources that overlap with the mainlobe

and/or sidelobe of the ship clutter, only eight relatively isolated ship clutter events were ultimately selected for the analysis of PSD statistics. For these clutter events, range gates where sidelobes were located were manually selected within 13.5 km (90 gates on Kumpula radar) in the radial direction and 15 degrees (15 rays) in the tangential direction, centered on the mainlobe. It is worth noting that the radius along radial and tangential directions are determined by the inherent characteristics of the pulse width and antenna pattern of Kumpula radar. For a pulse width of 90 μs (Table 1), the maximum

range affected by the range sidelobes of ship clutter is 13.5 km. Moreover, as the angle deviates from the mainlobe, the power of the sidelobe echoes decreases significantly. Therefore, the sidelobes beyond the 15-degree azimuthal range can be considered below the radar's sensitivity and thus unobservable. To facilitate statistical analysis across different ship clutter

events, the SNR values of the ship clutter were normalized (i.e., the $SNR_{main}$ and $SNR_{side}(\Delta x, \Delta y)$ were subtracted from the $SNR_{main}$ in dB units). Owing to several factors in actual observations, the relative relationship between the mainlobe and sidelobes of the eight ship clutter events was inconsistent. As previously discussed, to capture as many sidelobe distribution characteristics as possible (thereby ensuring the results are applicable across a wide range of scenarios), the maximum value of $SNR_{side}(\Delta x, \Delta y)$ from the eight ship clutter events was selected. Additionally, the maximum values of the antenna sidelobes were obtained on both sides, using the antenna mainlobe as the reference axis.

The statistical results of the relative power between the mainlobe and sidelobe are shown in Fig. 7. When a range gate was identified as the mainlobe of the ship clutter, the SNR differences between it and the surrounding range gates were calculated. Range gates with SNR differences exceeding the statistical results shown in Fig. 7 were identified as PSD.

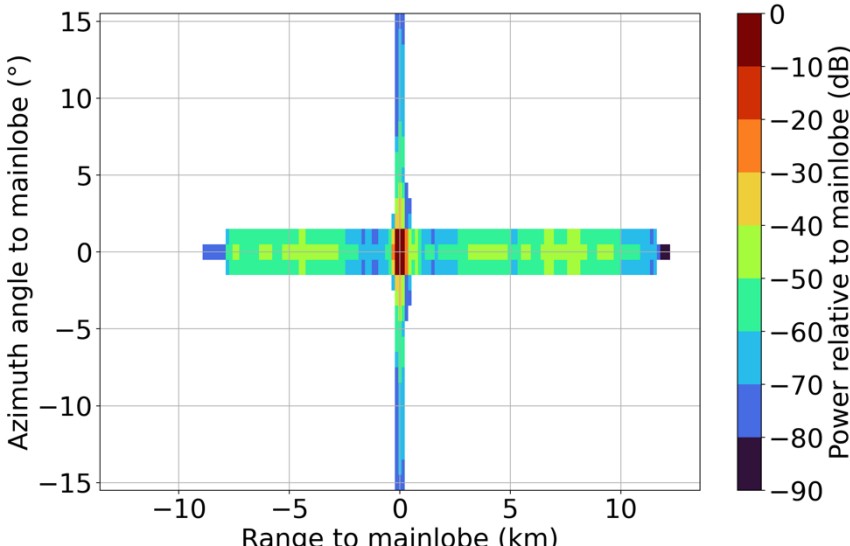

Figure 7: Statistical result of the relative power between the mainlobe and sidelobe for eight ship clutter events with high SNR.

### 3.2.2 Velocity and SNR filter

As shown in Fig. 8, the adaptively determined PSD is effective in identifying all affected range gates for isolated ship clutter. However, when ship clutter overlaps with other types of echoes, such as the precipitation echoes shown in Fig. 9, eliminating the PSD can lead to loss of important information. In other words, the PSD is a sufficient but unnecessary condition for the sidelobe distribution of the ship clutter. Thus, additional constraints are required to refine the PSD screening.

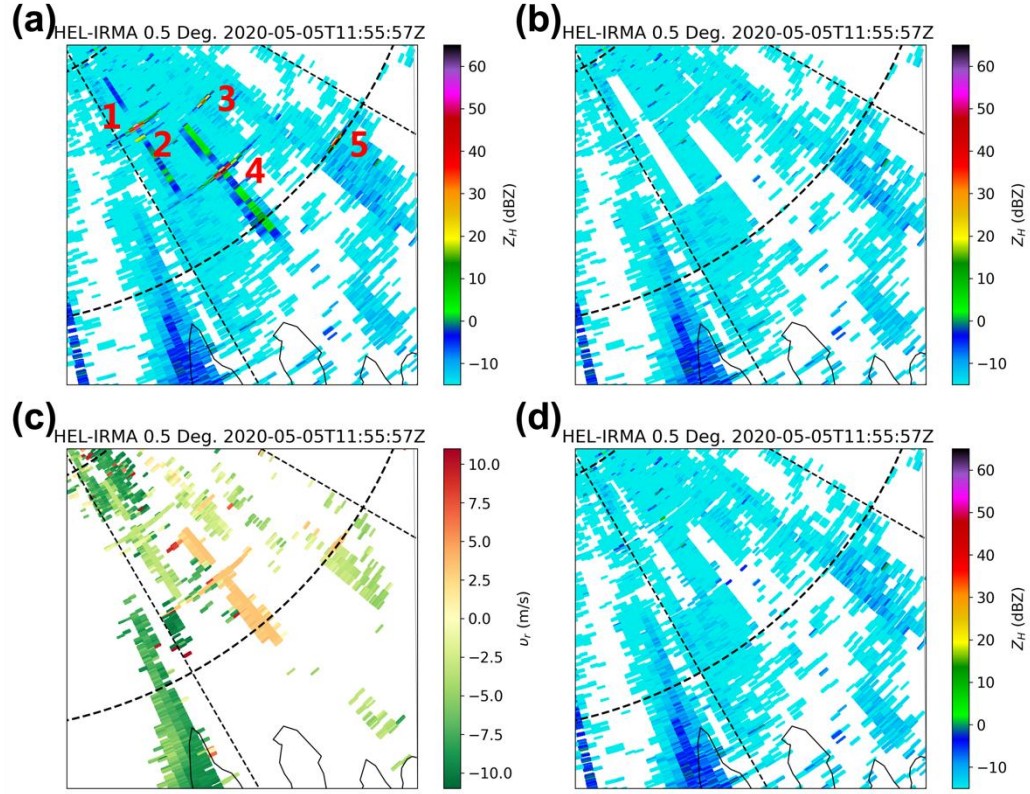

**Figure 8: 0.5° elevation of Kumpula radar using LFM waveform at 1155 UTC 5 May 2020. (a) Reflectivity before ship-clutter filtering; (b) Reflectivity after filtering all range gates in the PSD; (c) Doppler velocity; (d) Reflectivity after filtering using velocity and SNR thresholds.**

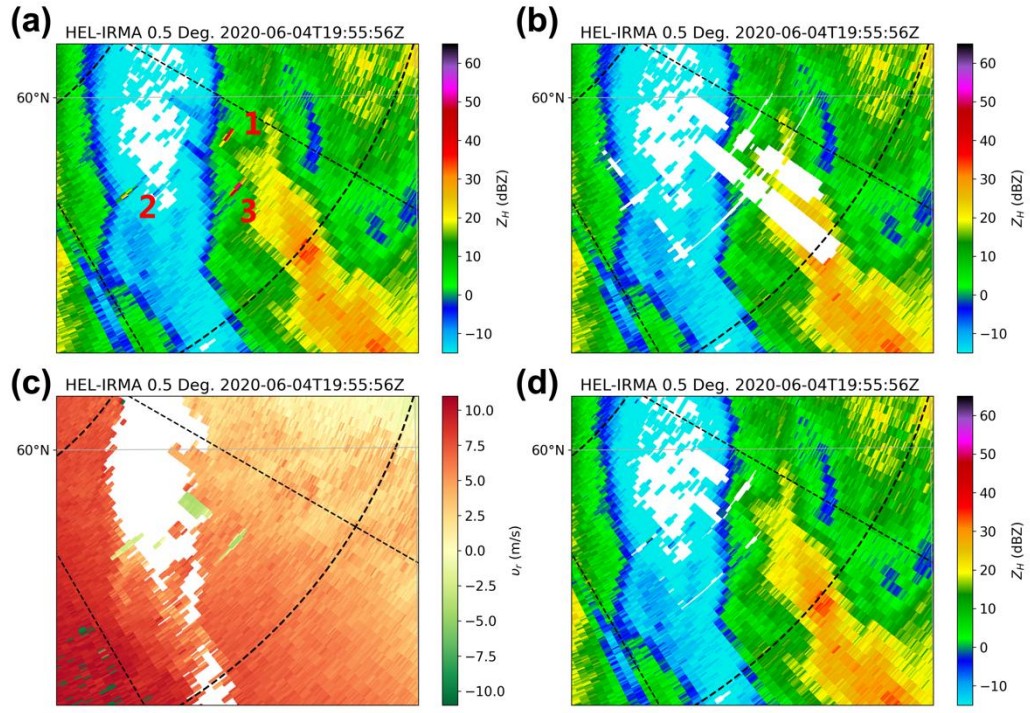

**Figure 9:0.5° elevation of Kumpula radar using LFM waveform at 1955 UTC 4 June 2020. (a) Reflectivity before ship-clutter filtering; (b) Reflectivity after filtering all range gates in the PSD; (c) Doppler velocity; (d) Reflectivity after filtering using velocity and SNR thresholds.**

When analyzing the signatures of ship clutter across different radar variables, it was found that the $v_r$ of the sidelobes exhibits consistent patterns. This consistency makes $v_r$ a highly effective indicator for PSD screening. As shown in Fig. 8a, among the five identified ship clutter events, Nos. 1 and 4 are particularly notable for their distinct cross-shaped patterns. Correspondingly, their $v_r$ (Fig. 8c) also display cross-shaped distributions with small differences within each group (standard deviations of 0.48 and 0.37 m/s, respectively).

To quantitatively assess the $v_r$ distribution of ship clutter, we selected datasets based on criteria like those for PSD statistics described in Sect. 3.2.1, albeit with less stringent SNR requirements. Consequently, statistics were gathered from 65 ship clutter samples. After normalizing the SNR and $v_r$ of the ship clutter sidelobes, the data was categorized into bins based on the power difference between the sidelobes and the mainlobe, ranging from -80 to 0 dB in 20 dB increments. The statistical outcomes, illustrated in the violin plot in Fig. 10, indicate that while there are outliers, a $v_r$ threshold of 1 m/s is adequate to encompass most ship clutter sidelobes.

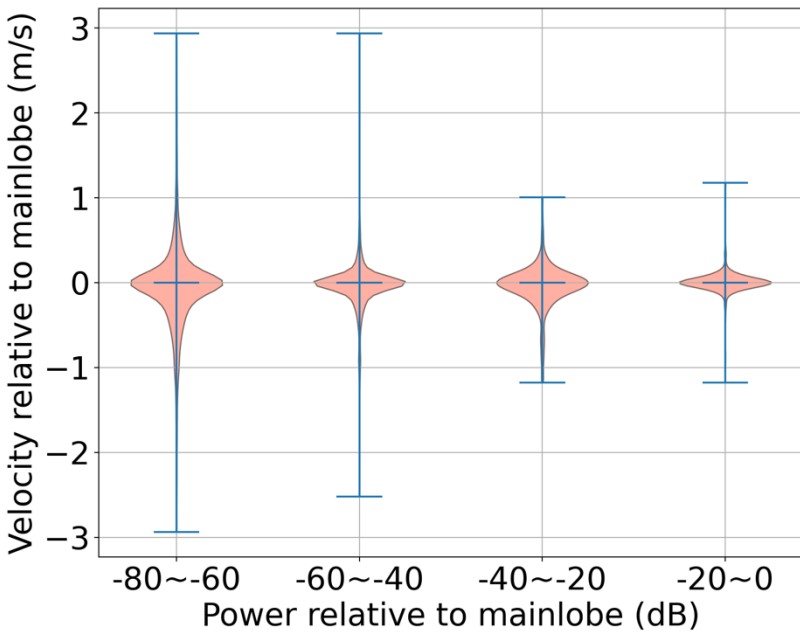

**Figure 10: Violin plots showing the normalizing $v_r$ for normalizing SNR of ship clutter sidelobes from -80 to 0 dB in steps of 20 dB.**

A notable exception, however, is ship clutter No. 1 in Fig. 9, where precipitation and ship clutter overlap and exhibit similar $v_r$ values. To address this, we introduced an additional SNR threshold, that is, the SNR of the ship clutter sidelobe must exceed the lower limit used for PSD determination in Sect. 3.2.1, but not exceed the higher SNR threshold (default to 10 dB).

Typically, the SNR of ship clutter sidelobes is lower than that of precipitation echoes.

The identification results, after applying $v_r$ and SNR thresholds, are presented in Figs. 8d and 9d. Compared with Fig. 9a, Fig. 9d effectively isolates and removes only the regions affected by ship clutter, with minimal loss of precipitation echo information. Meanwhile, there is no observable degradation in the identification performance for the isolated ship clutter between Figs 8a and 8d.

# 4 Performance evaluation

## 4.1 Case analysis

Figure 11a presents a typical clear air scenario observed by the Kumpula radar using LFM waveform at 1310 UTC on May 5, 2020, where several ship clutter events were observed in isolation from the precipitation and sea clutter. The results of the mainlobe identification are indicated by red circles. In Fig. 11a, which displays the $Z_H$ before the application of ship clutter

filtering, both strong cross-shaped and weaker point-shaped ship clutter are evident. Following the implementation of ship clutter filtering, as shown in Fig. 11b, both strong and weak ship clutter were effectively removed. The influence of ship clutter on $Z_H$ was substantially mitigated, rendering it visually undetectable.

Figure 11c presents the observation from the Kumpula radar at an adjacent time, using the NLFM waveform. To demonstrate the generalizability of the HSCI algorithm, we applied it to this case observed under the NLFM waveform. It is worth noting

that the mainlobe identification employed the same random forest model as that used for the LFM waveform. However, the relative power relationship between the mainlobe and sidelobes was re-estimated based on observations under the NLFM waveform, primarily due to the different sidelobe structures of the two waveforms. As shown in Fig. 11d, the HSCI algorithm achieves ship clutter suppression performance comparable to that under the LFM waveform. This demonstrates that the HSCI algorithm is not only effective for LFM waveforms but also exhibits good generalizability across different

waveform types.

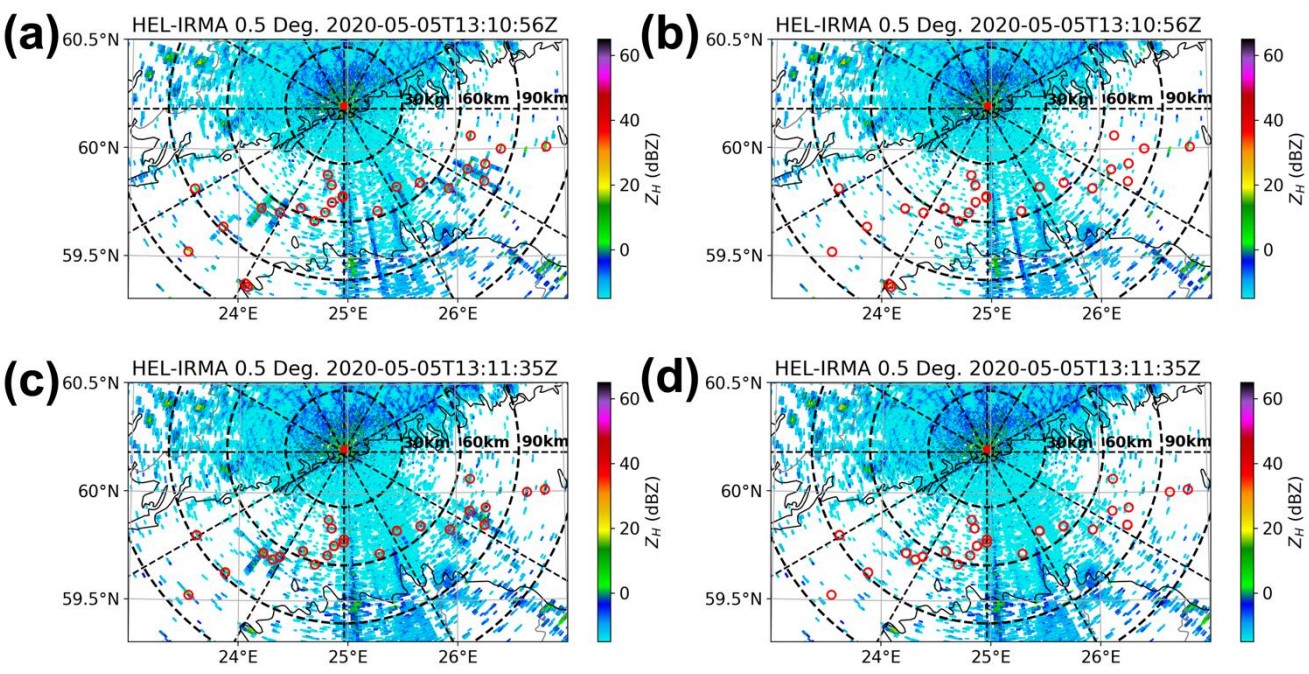

**Figure 11. Reflectivity on 0.5° elevation of Kumpula radar using LFM (1310 UTC) and NLFM (1311 UTC) waveforms on 5 May 2020. (a) Before ship clutter filtering using LFM waveform; (b) After ship clutter filtering using LFM waveform; (c) Before ship**
**clutter filtering using NLFM waveform; (d) After ship clutter filtering using NLFM waveform. The mainlobe identification results are highlighted by red circles.**

A precipitation event observed by the Kumpula radar at 2240 UTC on June 4, 2020 is shown in Fig. 12, in which several ship clutter events were completely mixed with precipitation echoes. In cases where ship clutter sidelobes are involved, the

echo intensity of most ship clutter sidelobes is generally lower than that of the adjacent precipitation echoes. As a result, the intrinsic characteristics of the ship clutter sidelobes, such as the $v_r$ values approaching those of the ship clutter mainlobe, are not prominently displayed in these overlapping range gates. Instead, the characteristics of the dominant precipitation echoes prevail. Therefore, the analysis primarily focuses on the identification of ship clutter mainlobes. This typical case includes the three main output scenarios of the HSCI algorithm: (1) ship clutter is successfully identified (highlighted with red or white circles); (2) precipitation echoes are mistakenly identified as ship clutter (red or white circles indicated by arrows); and (3) ship clutter is not effectively identified (highlighted with black rectangles). For the first scenario, a comparison between Figs. 12a and 12b shows that the impact of ship clutter has been effectively suppressed without compromising the surrounding precipitation echoes. In the second scenario, although errors occurred in the mainlobe identification, the application of velocity and SNR filters in sidelobe identification effectively controlled the loss of precipitation echoes—that is, only the range gate misidentified as a ship clutter mainlobe and its eight nearest neighboring gates were mistakenly removed. For the third scenario, although the presence of ship clutter can be inferred from polarimetric variables such as the $\rho_{HV}$ (Fig. 12d), its impact on $Z_H$ (Fig. 12a) is so minimal that it is virtually undetectable.

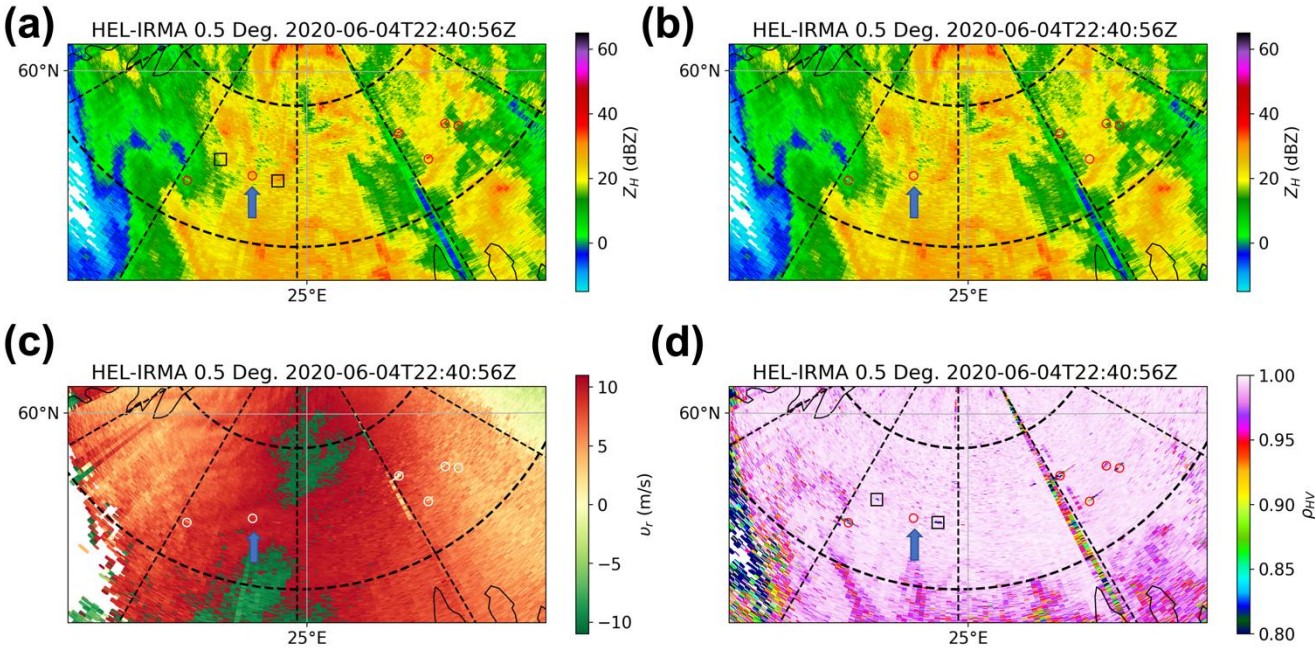

**Figure 12: 0.5° elevation of Kumpula radar using LFM waveform at 2240 UTC 4 June 2020. (a) Reflectivity before ship clutter filtering; (b) Reflectivity after ship clutter filtering; (c) Doppler velocity before ship clutter filtering; (d) co-polar correlation coefficient after ship clutter filtering. The mainlobe identification results are highlighted with red or white circles, with misidentified cases indicated by blue arrows, while the unrecognized ones are highlighted with black rectangles.**

## 4.2 Statistical evaluation

A quarter of the manually curated dataset, consisting of 400 gates for ship clutter and 2500 gates for precipitation echoes, was used to objectively assess the mainlobe identification results. The remainder of the dataset (75%) served as the training set for the random forest identification model. The three metrics—precision, recall, and F1-score—were used in this study to evaluate the performance of the HSCI algorithm. They are defined as follows:

$$\text{Precision} = \frac{TP}{TP+FP}, \tag{5}$$

$$\text{Recall} = \frac{TP}{TP+FN}, \tag{6}$$

$$F1-\text{score} = \frac{2\times\text{Precision}\times\text{Recall}}{\text{recision}+\text{Recall}}, \tag{7}$$

where TP denotes the number of range gates correctly identified, FP denotes the number of range gates incorrectly identified, and FN denotes the number of range gates were not identified. Table 4 summarizes the precision, recall and F1-score of the HSCI algorithm on the test dataset. For ship clutter, the algorithm achieved a precision of 99.49%, a recall of 97.25% and an F1-score of 98.36%. In the case of precipitation echoes, precision reached 99.56%, recall 99.92% and the F1-score 99.74%. These results demonstrate that the HSCI algorithm performs well on both echo types, with all three metrics exceeding 97%, and most exceeding 99%.

**Table 4: Performance evaluation results of HSCI algorithm for identifying ship clutter mainlobe based on the test dataset.**

| Echo type | Precision | Recall | F1-score |
|---|---|---|---|
| Ship clutter | 99.49% | 97.25% | 98.36% |
| Precipitation echoes | 99.56% | 99.92% | 99.74% |

The performance of the identification process was further quantified using a probability density plot like that shown in Fig. 6, where the overlapping area between the distributions of ship clutter and precipitation echoes was only 2.83%. This represents a significant improvement over the results obtained using a single feature, as detailed in Table 2, and underscores the benefits of integrating multiple features to enhance the identification accuracy.

In addition to using labeled datasets to evaluate the performance of mainlobe identification, this study also incorporated observed scanning data from the Kumpula radar. Unlike other studies that typically present identification results from one or a few radar scans (Tang et al., 2014; Kurdzo et al., 2020), our analysis encompasses a 24-hour precipitation event on June 4, 2020. The $Z_H$ both before and after the removal of ship clutter, was converted into precipitation rates ($R$) using the *Z-R* relationship $Z_H = 300R^{1.6}$ (Marshall and Palmer, 1948), and the total precipitation rate for the entire event was accumulated. The conversion of $Z_H$ to the precipitation rate has two important purposes. First, the precipitation rate is a critical parameter

in meteorological applications, offering a more direct reflection of identification performance. Secondly, it facilitates the accumulation of data, allowing for the analysis of long-term effects.

The rain accumulations before and after the removal of ship clutter are shown in Figs. 13a and 13b. As illustrated in Fig. 13a,
numerous areas with high precipitation accumulation appear in linear formations over the sea. Following the application of ship clutter removal, as shown in Fig. 13b, these anomalous values were effectively eliminated, whereas the precipitation echoes in other areas remained unaffected. This confirms the efficacy of the ship clutter identification algorithm and demonstrates its capability to enhance the accuracy of precipitation measurements. Figure 13c shows the difference in precipitation accumulation before and after ship clutter removal (i.e., subtracting the precipitation accumulation shown in Fig.
13b from that in Fig. 13a), highlighting the overestimation of precipitation caused by ship clutter. A comparison with the ship traffic density map of the Gulf of Finland shown in Fig. 13d reveals a strong correlation between the two, substantiating the assertion that these echoes originated from ships. Moreover, in the main navigation channel (the red area in Fig. 13d), the precipitation accumulation after ship clutter removal exhibits a smooth distribution similar to the surrounding areas. This suggests, to some extent, that the loss of precipitation echoes caused by the HSCI algorithm is negligible.


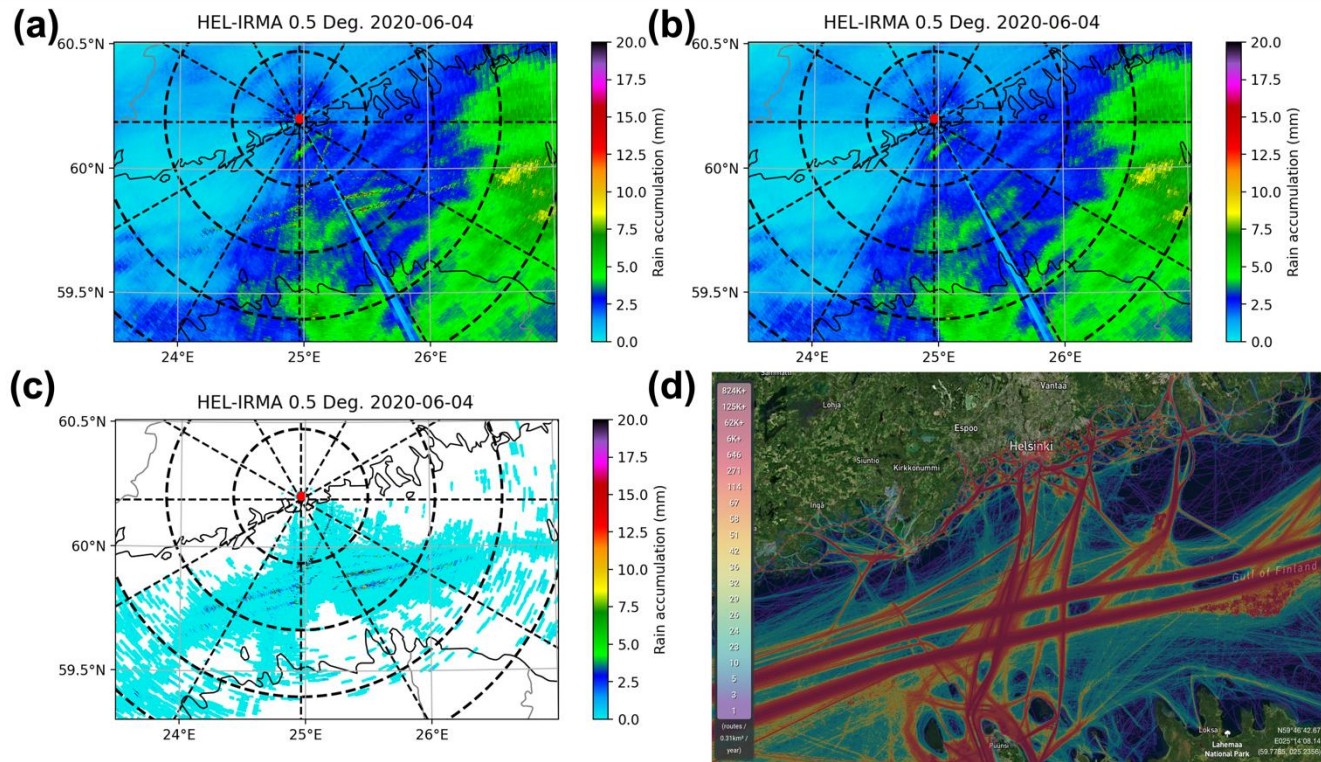

**Figure 13: 24-hour rain accumulation on 4 June 2020 from Kumpula radar at 0.5° elevation. (a) Before ship clutter filtering; (b) After ship clutter filtering; (c) Difference between before and after filtering (i.e., subtracting the precipitation accumulation shown in (b) from that in (a)); (d) Ship density map of the Gulf of Finland (© MarineTraffic 2024; Image source:**
**www.marinetraffic.com).**

Although a performance analysis based on a pre-constructed dataset has been conducted, a quantitative evaluation under real observational conditions is still lacking. The full-day observations from Kumpula radar on June 4 provide an opportunity to address this gap. We extracted 24 scans, selecting one scan per hour, and determined the locations of ship clutter using the same method described in Section 2. The evaluation results are summarized in Table 5. A total of 247 ship clutter events were identified across the 24 scans (i.e., the sum of the second and third columns in the second row). Of these, the HSCI algorithm successfully detected 238, missing only 9. Additionally, 31 range gates associated with precipitation echoes were mistakenly identified as ship clutter. However, as discussed in Section 4.1, the actual loss of precipitation echoes is well controlled, thanks to the velocity and SNR thresholds integrated into the HSCI algorithm.

**Table 5: Performance evaluation results of HSCI algorithm for identifying ship clutter mainlobe based on observations from 4 June 2020.**

| Time (UTC) | Hit | Miss | False Alarm |
|:---:|:---:|:---:|:---:|
| Total | 238 | 9 | 31 |
| 00:00:41 | 21 | 0 | 2 |
| 01:00:40 | 14 | 0 | 2 |
| 02:00:40 | 10 | 1 | 4 |
| 03:00:41 | 11 | 1 | 7 |
| 04:00:40 | 14 | 0 | 2 |
| 05:00:40 | 12 | 1 | 0 |
| 06:00:41 | 8 | 0 | 1 |
| 07:00:40 | 9 | 0 | 2 |
| 08:00:40 | 7 | 1 | 0 |
| 09:00:41 | 4 | 0 | 0 |
| 10:00:41 | 2 | 0 | 1 |
| 11:00:41 | 6 | 1 | 1 |
| 12:00:41 | 10 | 1 | 0 |
| 13:00:41 | 16 | 0 | 3 |
| 14:00:40 | 14 | 0 | 2 |
| 15:00:40 | 16 | 0 | 0 |
| 16:00:40 | 11 | 0 | 0 |
| 17:00:41 | 8 | 1 | 0 |
| 18:00:41 | 12 | 0 | 2 |
| 19:00:40 | 10 | 0 | 1 |

| 20:00:41 | 11 | 0 | 0 |
|---|---|---|---|
| 21:00:41 | 4 | 0 | 0 |
| 22:00:41 | 6 | 1 | 1 |
| 23:00:40 | 2 | 1 | 0 |

## 5. Discussion

The evaluation results confirm that the HSCI algorithm effectively identifies both the mainlobe and sidelobes of ship clutter observed by Kumpula radar. Combined with its logical simplicity and high computational efficiency, it appears highly promising for implementation in commercial signal processors (e.g., RVP900). However, due to the cross-radial effect of the antenna sidelobes, the implementation of the HSCI algorithm requires a buffering delay of several hundred milliseconds for the data from these radials, limiting its performance to quasi-real-time operation. Nonetheless, this represents a cost-effective

compromise for reducing the influence of ship clutter on radar data. Additionally, some configuration information needs to be pre-integrated into the signal processor. For example, geographic information about the sea area must be collected in advance based on the radar deployment location and used in the region limitation step of the HSCI algorithm. In addition, other radars may have different antenna patterns and pulse compression waveforms compared to the Kumpula radar, which makes the statistical results obtained in this study inapplicable to them. Therefore, when applying the HSCI algorithm to

other radars, a large amount of ship clutter data must be manually selected for algorithm training and characteristics analysis, which is a labor-intensive yet necessary task.

    In this study, the method used to suppress the negative impact of ship clutter is to mask radar variables at the range gates where ship clutter is identified. However, when precipitation echoes overlap with ship clutter (as shown in Fig. 12), this inevitably leads to the loss of precipitation data. An effective strategy to address this challenge is to move the stage of ship

clutter identification and filtering from "data processing" to "signal processing" (Keeler and Passarelli, 1990), which will become feasible when the HSCI algorithm is integrated within the signal processor. This approach involves using spectrum processing techniques to suppress the ship clutter component in the radar signal while preserving the precipitation component like ground clutter filtering methods such as GMAP (Siggia and Passarelli, 2004) and CLEAN-AP (Torres and Warde, 2014). The center position of the notch filter needs to be adjusted from zero frequency to match the $v_r$ of the ship

clutter's mainlobe.

    During the evaluation of the HSCI algorithm (Section 4.1), we observed that weak ship clutter affects overlapping precipitation echoes differently in reflectivity and polarimetric variables, with the latter being more susceptible to contamination. In such cases, a general-purpose non-meteorological echo filter based on a correlation coefficient threshold could be used as an optional supplement to the HSCI algorithm in operational settings (Tang et al., 2014), helping to mitigate

the potential impact of residual weak ship clutter on meteorological applications. In addition, the evaluation revealed that the

HSCI algorithm has a tendency to over-identify ship clutter. However, since the algorithm employs velocity and SNR filters in the sidelobe identification, even when precipitation echoes are mistakenly identified as ship clutter, the resulting precipitation loss can be kept to a minimum. For these limited range gates that are incorrectly removed, interpolation techniques or speckle filters can effectively restore the missing information (Vaisala, 2016).

Due to the limitations of available observational data, this study develops and evaluates the HSCI algorithm primarily based on ship clutter observed by the Kumpula radar in the Gulf of Finland, using an LFM waveform. While the features designed for ship clutter identification and the overall logic of the HSCI algorithm are intended to be general and not tied to a specific radar system or waveform, we acknowledge that the current validation is restricted to a single radar platform, geographic location, sea state, and waveform configuration. Therefore, the robustness and generalizability of the algorithm under

varying conditions—such as across different radar systems, geographical regions, environmental conditions (e.g., sea clutter dynamics influenced by wind or temperature), seasons, and transmitted waveforms—remain to be comprehensively assessed. We recognize this as a significant limitation of the present work. Future studies will aim to address this issue by applying the HSCI algorithm to a broader range of observational datasets, enabling a more rigorous evaluation of its performance and adaptability in diverse operational scenarios.

**6. Summary**

In this study, a Hybrid Ship Clutter Identification (HSCI) algorithm for pulse compression weather radar was introduced. This algorithm not only identifies the mainlobe of the ship clutter but also detects the range sidelobes resulting from pulse compression technology and the antenna sidelobes inherent to all radars. Data observed using the Kumpula radar at the University of Helsinki (from May to June 2020), which frequently captures the activities of ships sailing in the Gulf of

Finland, were used in this study. For algorithm development and performance evaluation, 1,600 ship clutter and 10,000 range gates of precipitation echoes were manually selected.

The HSCI algorithm is structured into two parts: mainlobe and sidelobe identification. In the mainlobe identification step, the identification region is initially limited to minimize false identifications and enhance efficiency. Six features—RD, RGF, $\sigma_v$, SWR, $Z_{DR}$, and CCD—are then calculated and used in a random forest model to distinguish ship clutter mainlobe from

precipitation echoes. If the model confirms the presence of a ship clutter mainlobe, the process transitions to sidelobe identification.

The potential sidelobe distribution (PSD) of ship clutter is dynamic, increasing or decreasing with the SNR of the mainlobe. The first step in sidelobe identification uses an adaptive method to accurately determine the PSD, thus avoiding missed identifications that leave sidelobe residues or excessive identifications that lead to precipitation data loss. Velocity and SNR

filters were then applied within the PSD to further protect against the loss of precipitation information due to overlapping ship clutter and precipitation echoes.

Two typical cases (one in clear air and the other during precipitation) were used for the algorithm performance analysis. The results demonstrate that isolated ship clutter is accurately identified and filtered out, whereas ship clutter overlapping with precipitation is also effectively identified and removed while preserving the precipitation data. Even in cases where precipitation is misidentified as ship clutter, the loss of actual precipitation echoes is minimized due to the constraint mechanisms embedded in the HSCI algorithm. In addition, the algorithm (mainlobe identification part) was evaluated on a test dataset comprising 400 ship clutter gates and 2,500 range gates of precipitation echoes. For ship clutter, the algorithm achieved a precision of 99.49%, a recall of 97.25% and an F1-score of 98.36%. In the case of precipitation echoes, precision reached 99.56%, recall 99.92% and the F1-score 99.74%. This study also evaluated the cumulative precipitation before and after ship clutter filtering during a 24-hour precipitation event. The results show that the precipitation overestimation caused by ship clutter was effectively eliminated, and the footprint of precipitation overestimation corresponded well with the ship density map. Furthermore, the quantitative evaluation based on the 24-hour observations (rather than the pre-extracted dataset) also demonstrate the robust performance of the HSCI algorithm.

It is worth mentioning that the phased array technology is becoming increasingly prevalent in weather radar. Since the early 21st century, the United States has conducted experiments and developed advanced phased array weather radars such as ATD (Weber et al., 2021) and Horus (Palmer et al., 2023). Similarly, China pioneered the operational application of phased array technology in several provinces (Geng and Liu, 2023). Pulse compression technology, a cornerstone of active phased array radars, suggests that the HSCI algorithm proposed in this study may have even wider applications in the future. Moreover, phased array radars often use digital beamforming for rapid scanning. However, this can exacerbate the deterioration of the antenna sidelobes in the direction of elevation (Schvartzman et al., 2021). Consequently, it is anticipated that future HSCI algorithms will evolve from two-dimensional to three-dimensional.

**Appendix**

SNR is commonly used as a threshold parameter to mask regions with noise and weak signals that are significantly influenced by noise. Although SNR can be included as part of the archived data along with other radar parameters such as $Z_H$, $v_r$, $\sigma_v$, $Z_{DR}$, $\phi_{DP}$, and $\rho_{HV}$, in some signal processors, it is not always mandatory. For instance, the Kumpula radar used in this study did not output an SNR. Given that the SNR is a crucial factor for adaptively determining the PSD in the HSCI algorithm, an estimation technique was developed to accurately obtain the SNR using reflectivity ($Z$).

In Vaisala RVP signal processors, $Z$ (expressed in logarithmic units) is estimated from the SNR (expressed in logarithmic units) and a series of constants (Vaisala, 2016):

$$Z = SNR + C .\tag{A1}$$

For simplicity, a series of constants is consolidated and denoted by $C$. When the SNR reached the preset threshold $SNR_{thr}$ (1.5 dB for the Kumpula radar), the radar detected the minimum reflectivity $Z_{min}$:

$$Z_{min} = SNR_{thr} + C .\tag{A2}$$

Subtracting Eq. (A2) from Eq. (A1):

$$Z - Z_{min} = SNR - SNR_{thr} .\tag{A3}$$

The SNR can then be determined by transposing:

$$SNR = Z - Z_{min} + SNR_{thr} .\tag{A4}$$

In Eq. (A4), $Z$ can be sourced from the archived data, and $SNR_{thr}$ is a predefined value set by the user. $Z_{min}$ for each range can be determined through statistical analysis of a large dataset.

It is worth emphasizing that the proposed SNR estimation method can be viewed as the inverse of the reflectivity estimation. As such, the derived SNR maintains the same level of accuracy as that obtained directly from the raw time series data, without introducing any additional errors.

## Data availability

The radar data used in this study are available from Leskinen Matti or Dmitri Moisseev upon request.

## 540 Author contributions

S.Z. designed the algorithm details, developed the algorithm code, and drafted the initial manuscript. H.L. revised and polished the manuscript. D.M. provided critical guidance on algorithm design. L.M. curated the dataset required for algorithm development.

## Competing interests

The contact author has declared that none of the authors has any competing interests.

## Acknowledgements

The authors wish to express their gratitude to the editor and the reviewers for the constructive and positive comments on this work.

## Financial support

This research has been supported by the National Natural Science Foundation of China (grant no. 42405141) and Open Foundation of the China Meteorological Administration Tornado Key Laboratory (grant no. TKL202310).

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
