# Peer review of "A hybrid algorithm for ship clutter identification in pulse compression polarimetric radar observations"

_Atmospheric Measurement Techniques, 2024_

## Referee Comment (RC2)

**The paper presents a novel method for detecting the mainlobe of ship clutter using multiple features and machine learning, instead of relying on a single parameter. While the manuscript is well-organized and generally easy to follow, it contains several significant issues that should be addressed before publication.**

**major revisions:**

- **Precipitation Dataset Clarification (Line 110):**

The manuscript mentions that the precipitation dataset was manually extracted. However, it is unclear why this was necessary, especially since the algorithm is intended to work under both clear-sky and rainy conditions. If the algorithm is designed to perform differently depending on weather conditions, this distinction should be clearly stated and justified in the paper.

- **Threshold Justification (Line 136):**

The choice of a 20 dBZ threshold is not explained. Where does this value come from? Additionally, does ship clutter with lower reflectivity (e.g., 15 dBZ) not affect data quality? This requires clarification.

- **Manual Selection of Sidelobe Region (Line 244):**

The sidelobe region is manually defined within a 13.5 km radial range and 15° in the tangential direction. The origin and justification for these values are not provided. Were these same values used later in operational scenarios? Furthermore, what automated method replaces the manual process in operational applications?

- **Case Study Placement (Section 3):**

Two case studies are introduced in the ‚method' section. However, they seem more appropriate for Section 4, where other case analyses are presented. Additionally, the purpose of second case (Fig. 8) is unclear—what characteristics is it meant to highlight, and why was it chosen?

- **Discussion Section Issues:**

The final paragraph of the discussion section appears to be more suitable for the introduction or summary. Once this paragraph is removed, the discussion becomes quite short. It would benefit from more in-depth coverage of how the algorithm operates in operational case—especially regarding the parts that were handled manually (e.g., sidelobe removal). Does the algorithm only remove sidelobes near the mainlobe, or is there a more general solution? These points should be explicitly addressed.

- **Operational Performance for Winter Events:**

Since the algorithm is intended for operational use, it is strongly recommended to include at least one winter precipitation case to demonstrate its robustness in varying weather conditions.

**minor correction:**

- Figure 3 is explained in the introduction but appears on page 4. Consider repositioning it closer to the relevant text.
- In several instances, figures are referenced before they are introduced or explained (e.g., Line 321: *"This selective filtering approach is demonstrated in Figs. 11b and 12d…").* However, Figure 12 is not described until Line 345. This disrupts the flow and may confuse readers. Please revise for consistency.
- Fig 12c: it is not clear if the plot shows the difference ‚before-after' filtering or vise versa

---

## Author Response (AR1)

**Response to Reviewer 1**

**Question 1:** Lack of Rigorous Validation Metrics

The model evaluation focuses on overall accuracy and a small overlap percentage in histograms. However, these metrics are insufficient for a classification task with imbalanced classes (e.g., 400 vs. 2,500 gates in the test set). The manuscript should report precision, recall, and F1-score, especially for ship clutter, as false negatives can lead to significant data quality issues, and false positives can unnecessarily degrade precipitation data.

*Agree. In the revised manuscript, the additional evaluation metrics suggested by the reviewer have been included in Section 4.2* (Line 360; highlighted content). *We have also incorporated these metrics into the abstract and summary have also been revised* (Line 19 and 450; highlighted content).

**Question 2:** Limited Generalization and Dataset Diversity

The random forest model is trained and tested on data derived from the same radar (Kumpula), location (Gulf of Finland), waveform (LFM), and limited events. There is no evidence that the algorithm generalizes to other waveform types (e.g., NLFM), elevation angles, or environmental conditions (e.g., high sea clutter, near-shore echoes, different clutter types). The authors must either test the model on independent cases or clearly state the generalization limitations.

Agree. This is a pilot study on identifying ship signatures in polarimetric observations, and we may not overstate its potential general applications. Although we believe that the features and the algorithmic logic proposed for identifying ship clutter are generally applicable, the manuscript lacks an evaluation of the algorithm's robustness across multiple scenarios (e.g., different radars, sea areas, waveforms, etc.). To address this issue, we have added the discussion of its limitations in Section 5 (Line 425; highlighted content) and more in-depth analysis should be made in future.

**Question 3:** Overreliance on Manual Labeling

Both the ship clutter and precipitation echo datasets are manually labeled, and the methodology for doing so is not sufficiently described. This introduces potential biases. What criteria were used to define clutter? Were multiple annotators involved? Was any inter-annotator agreement measured? These questions should be addressed or acknowledged as limitations.

We feel sorry for the unclear description in the previous version. The datasets were labelled by the first author, and the validity of ship clutter identification is assured by checking its movement continuity in consecutive radar scans. We acknowledge that an objective approach could be more comparable to future studies, but no such methods are available. In the revised manuscript, we have supplemented the relevant explanations and figure in Section 2 (Line 112 and 134; highlighted content).

**Question 4:** Sidelobe Suppression Logic May Be Overly Aggressive

The PSD definition and filtering logic—especially the combination of velocity and SNR thresholds—may lead to over-removal of precipitation echoes, particularly in overlap regions. While some case studies suggest selective filtering is achieved, the possibility of precipitation loss is real and must be quantified. For example, is there a statistical estimate of how many precipitation gates were removed in mixed scenes?

According to the current ship clutter identification and removal method presented in this paper, when precipitation overlaps with ship clutter, the precipitation information within the same range gate will be lost if the ship clutter is successfully identified (as mentioned in the Discussion, this issue needs to be addressed through signal-level processing). The HSCI algorithm employs velocity and SNR filters with the aim of minimizing the loss of precipitation information. However, the authors acknowledge that there is still a risk of over-suppression even with these measures.

Since the algorithm proposed in this paper is based on data-level processing, a truly quantitative evaluation of precipitation loss would require signal-level techniques—specifically, the construction of ground truth fields through either full signal simulation (Curtis 2025) or semi-simulation (Li et al., 2013). Although we have already acquired the relevant technical capabilities, discussing these in the current paper would risk either deviating from the main topic (due to excessive description of signal processing techniques) or lacking sufficient detail. Therefore, we are preparing a manuscript, focusing on ship clutter suppression from the signal processing perspective to address this evaluation.

While a quantitative assessment of precipitation loss is not provided in this manuscript, Figure 13b offers a qualitative perspective: after suppressing the impact of ship clutter, the accumulated precipitation field

exhibits a smooth and natural distribution. Moreover, no abnormally low values are observed in the main navigation channel area (Figure 13d), indirectly suggesting that the HSCI algorithm does not cause detectable damage to precipitation echoes. Related analysis on precipitation loss has been added to Sections 4.2 of the revised manuscript (Line 394; highlighted content).

Li Y, Zhang G, Doviak R J, et al. A new approach to detect ground clutter mixed with weather signals[J]. IEEE Transactions on Geoscience and Remote Sensing, 2012, 51(4): 2373-2387.

Curtis C D. Weather Radar Time Series Simulation: Rapidly Looping through Signal Parameters[J]. Journal of Atmospheric and Oceanic Technology, 2025, 42(3): 269-279.

**Question 5:** No Independent Test Dataset or Cross-validation

The authors should demonstrate model robustness through k-fold cross-validation or by holding out an entire day or event as an independent test case. Without this, it is difficult to assess whether the model is overfitting or merely capturing spatiotemporal autocorrelation patterns.

The author sincerely apologizes for not explaining this issue clearly in the original manuscript. In Section 3.1.3, we mention the use of the GridSearchCV method from Scikit-learn for optimizing model parameters. This method combines exhaustive search and cross-validation to determine the optimal parameters. The cross-validation referred to here is exactly the k-fold cross-validation suggested by the reviewer (with the default value of k being 5). Additional details have been added to Section 3.1.3 in the revised manuscript (Line 237; highlighted content).

**Question 6:** Lack of Code or Reproducibility Path

For a method combining machine learning and empirical filtering, reproducibility is essential. At a minimum, a flowchart covering the entire algorithmic sequence, and pseudocode or a link to a repository, should be provided. Currently, implementation details are scattered and would be difficult for others to reproduce.

Good suggestion. In our resubmission, we have included the relevant code and data. By combining the code with the flowchart shown in Fig. 5, we believe that readers will find it easier to understand and master the algorithm.

**Question 7:** Feature Descriptions (Sect. 3.1.2): Clarify if all features are used as-is or normalized/scaled. Include distributions (box plots or ranges) for ship vs. precip echoes for added transparency.

Agree. We have supplemented the relevant explanations in Section 3.1.2 (Line 207; highlighted content).

**Question 8:** Figures 6 and 9: Use larger labels and more distinct color schemes. Consider adding contour lines for better interpretability.

Agree. The revised manuscript has optimized the figures (Fig. 7 and 10 now) as requested by the reviewer.

**Question 9:** Discussion Section: Expand on the feasibility of signal-level suppression in commercial processors (e.g., how would this be integrated in RVP900 or similar systems?). A brief note on real-time considerations would also help.

Agree. We have supplemented the relevant explanations in Section 5 (Line 405; highlighted content).

**Question 10:** Line 134: "efficiency is exceedingly low" → consider "computational efficiency is low"

Agree.

**Question 11:** Line 373: Remove extra parenthesis in "Palmer et al., 2023))".

Agree.

**Response to Reviewer 2**

**Question 1:** Precipitation Dataset Clarification (Line 110): The manuscript mentions that the precipitation dataset was manually extracted. However, it is unclear why this was necessary, especially since the algorithm is intended to work under both clear-sky and rainy conditions. If the algorithm is designed to perform differently depending on weather conditions, this distinction should be clearly stated and justified in the paper.

Weather radars are primarily used to observe precipitation echoes. However, in addition to precipitation, they also detect non-meteorological echoes such as ground clutter, biological echoes, and ship clutter, which is the focus of this study. These non-meteorological echoes can interfere with the observation and application of precipitation data, and therefore often need to be removed. As mentioned in Introduction, ground clutter and biological echoes are common types of non-meteorological echoes, and mature techniques have already been developed to suppress them effectively. In contrast, ship clutter is a less studied type of non-meteorological echo, and suppression methods for it have not yet been established—this is the primary research focus of our study.

Given that ground clutter and biological echoes can be effectively suppressed, we assume they have minimal or negligible impact on our research. Therefore, the classification task in this study is simplified to a binary classification problem between ship clutter and precipitation echoes, which is a common practice in the weather radar field (Kilambi et al., 2018; Hubbert et al., 2009).

When applying machine learning to solve a binary classification problem, it is necessary to identify features that can distinguish between the two classes and train a model to learn the feature distribution of each. This process requires a large amount of labeled data (i.e., with known ground truth), which explains why manual extraction of precipitation data is needed. In short, the reason for manually extracting precipitation data is not because the algorithm depends on specific weather conditions, but because the problem is formulated as a binary classification task, which requires a dataset for model training. Further clarification on this point has been added in Introduction of the manuscript (Line 83; highlighted content).

Hubbert J C, Dixon M, Ellis S M. Weather radar ground clutter. Part II: Real-time identification and filtering[J]. Journal of Atmospheric and Oceanic Technology, 2009, 26(7): 1181-1197.

Kilambi A, Fabry F, Meunier V. A simple and effective method for separating meteorological from nonmeteorological targets using dual-polarization data[J]. Journal of Atmospheric and Oceanic Technology, 2018, 35(7): 1415-1424.

**Question 2:** Threshold Justification (Line 136): The choice of a 20 dBZ threshold is not explained. Where does this value come from? Additionally, does ship clutter with lower reflectivity (e.g., 15 dBZ) not affect data quality? This requires clarification.

For pulse compression radars, in addition to the range gate of the ship target itself, ship clutter may also be observed in surrounding areas due to range sidelobes and antenna sidelobes. However, due to the inherent sensitivity limitations of radars, weaker sidelobe returns from ship clutter are often undetectable. Taking the high-sensitivity Kumpula radar as an example, the minimum detectable reflectivity at a range of 50 km is approximately -20 dBZ. As shown in the statistical results in Fig. 7, when a ship clutter with a mainlobe reflectivity of 20 dBZ is observed at 50 km, detectable sidelobe echoes are limited to a few range gates near the mainlobe (i.e., those exceeding -40 dB in Fig. 7). Namely, no significant sidelobes are observed by the radar if a lower threshold is used. Therefore, the selection of 20 dBZ is a comprise between radar sensitivity and the sidelobe levels. If better sensitivity can be achieved, a lower threshold can be made.

A more intuitive example is the ship clutter labeled as "2" in Fig. 9a. In such cases, when the ship clutter appears in isolation (i.e., not overlapping with precipitation echoes), it can be effectively suppressed using methods commonly applied to mitigate point clutter (Vaisala 2016). When the ship clutter overlaps with precipitation, the impact on the observation data is usually minimal. Therefore, by applying this threshold, the efficiency of the algorithm can be improved without affecting its performance. Additional explanation regarding this has been included in Section 3.1.1 of the manuscript (Line 154; highlighted content).

Vaisala: User's manual: RVP900 digital receiver and signal processor; Vaisala oyj: Vantaa, Finland, 513, 2016.

**Question 3:** Manual Selection of Sidelobe Region (Line 244): The sidelobe region is manually defined within a 13.5 km radial range and 15° in the tangential direction. The origin and justification for these values are not provided. Were these same values used later in operational scenarios? Furthermore, what automated method replaces the manual process in operational applications?

The radial range of 13.5 km (90 range gates) was determined based on the pulse width (90 μs) of the pulse compression waveform. In other words, the maximum range affected by the range sidelobes of ship clutter is 13.5 km. The tangential direction was limited to 15 degrees, based on the antenna pattern. As the angle deviates from the mainlobe, the power of sidelobe echoes significantly decreases. As shown in Fig. 7, when the deviation from the mainlobe reaches 15 degrees, the sidelobe level is 80 dB lower than the mainlobe. Therefore, data beyond this 15-degree range can be considered below the radar's sensitivity and thus undetectable. Additional explanation regarding this has been included in Section 3.2.1 of the manuscript (Line 270; highlighted content).

Regarding the reviewer's question about the manual data selection, this was performed during the algorithm design phase. The purpose was to analyze the characteristics of ship clutter under a specific radar (the Kumpula radar in this case) and a specific transmitted waveform (the 90 μs LFM waveform used by the Kumpula radar). In the subsequent application phase, manual data selection is no longer involved; instead, the statistical results obtained during the design phase are used. It is worth noting that the characteristics obtained in this study are limited to the Kumpula radar under the LFM waveform. When applying the algorithm to other radars, a new statistical analysis must be conducted.

**Question 4:** Case Study Placement (Section 3): Two case studies are introduced in the ‚method' section. However, they seem more appropriate for Section 4, where other case analyses are presented. Additionally, the purpose of second case (Fig. 8) is unclear—what characteristics is it meant to highlight, and why was it chosen?

This concern is valid. We did give thinking on this point in previous submission. However, considering that the readers of this paper may not all be experts in the field, relying solely on textual descriptions might not be reader-friendly. Therefore, following the writing style of Tang et al. (2014, 2020), we adopted a visual approach to present the algorithm's principles and design logic, which we believe will better aid readers in understanding the proposed method.

Secondly, the reason of presenting the second case is to highlight the effectiveness of the velocity and SNR filters in preventing excessive suppression of precipitation information. Specifically, under clear-sky conditions (i.e., the first case), since there is no precipitation echo, there is no significant difference whether the velocity and SNR filters are applied (Figs. 8b and 8d); in contrast, when ship clutter and precipitation echoes are mixed (i.e., the second case), using the velocity and SNR filters can greatly reduce the algorithm's loss of precipitation echoes (Figs. 9b and 9d).

Tang, L., J. Zhang, C. Langston, J. Krause, K. Howard, and V. Lakshmanan, 2014: A Physically Based Precipitation–Nonprecipitation Radar Echo Classifier Using Polarimetric and Environmental Data in a Real-Time National System. Wea. Forecasting, 29, 1106–1119

Tang, L., J. Zhang, M. Simpson, A. Arthur, H. Grams, Y. Wang, and C. Langston, 2020: Updates on the Radar Data Quality Control in the MRMS Quantitative Precipitation Estimation System. J. Atmos. Oceanic Technol., 37, 1521–1537

**Question 5:** Discussion Section Issues: The final paragraph of the discussion section appears to be more suitable for the introduction or summary. Once this paragraph is removed, the discussion becomes quite short. It would benefit from more in-depth coverage of how the algorithm operates in operational case—especially regarding the parts that were handled manually (e.g., sidelobe removal). Does the algorithm only remove sidelobes near the mainlobe, or is there a more general solution? These points should be explicitly addressed.

Agree. In the revised manuscript, the final paragraph of the Discussion section has been moved to the Summary, and the Discussion has been further improved (Line 456).

However, we would like to further clarify to the reviewer that, as described in Section 3.2.1 of the manuscript, the determination of the PSD is an adaptive process. Specifically, for strong ship clutter, the PSD closely matches the statistical results shown in Fig. 7. For weak ship clutter, however, the PSD is constrained by the radar's sensitivity — most sidelobes cannot be detected by the radar, resulting in a significantly smaller range limited to a few range gates adjacent to the mainlobe.

**Question 6:** Operational Performance for Winter Events: Since the algorithm is intended for operational use, it is strongly recommended to include at least one winter precipitation case to demonstrate its robustness in varying weather conditions.

This is a good suggestion. The occurrence of super-refraction which is conductive to ship clutter signatures has seasonal dependence (Lopez, 2009). Usually, ship clutter signatures are more common in summer than in winter. We have looked through our observations in the winter of 2020, but identified very limited ship clutter cases in precipitation. We found a single case where ship clutter was mixed with

precipitation, with just one ship clutter present across the entire PPI. Fortunately, the HSCI algorithm successfully detected and filtered this ship clutter without excessively suppressing the surrounding precipitation echoes. Further clarification on this point has been added in DATA section of the manuscript (Line 112; highlighted content).

Lopez, P. (2009). A 5-yr 40-km-resolution global climatology of superrefraction for ground-based weather radars. Journal of applied meteorology and climatology, 48(1), 89-110.

[Figure]

**Figure 1: 0.5° elevation of Kumpula radar using LFM waveform at 0010 UTC 6 Jan 2020. (a) Reflectivity before ship clutter filtering; (b) Reflectivity after ship clutter filtering; (c) Differential reflectivity before ship clutter filtering; (d) Differential reflectivity after ship clutter filtering; (e) Correlation coefficient before ship clutter filtering; (f) Correlation coefficient after ship clutter filtering.**

**Question 7:** Figure 3 is explained in the introduction but appears on page 4. Consider repositioning it closer to the relevant text.

Agree. The revised manuscript has been updated according to the reviewer's suggestion to move Figure 3 forward.

**Question 8:** In several instances, figures are referenced before they are introduced or explained (e.g., Line 321: "This selective filtering approach is demonstrated in Figs. 11b and 12d…"). However, Figure 12 is not described until Line 345. This disrupts the flow and may confuse readers. Please revise for consistency.

We mistakenly wrote 11d as 12d, which would indeed mislead readers. We are very grateful to the reviewer for their careful review and for pointing out this error on our behalf.

**Question 9:** Fig 12c: it is not clear if the plot shows the difference ‚before-after' filtering or vise versa

Agree. In the revised manuscript, we have supplemented the relevant explanations in Section 4.2 (Line 389; highlighted content).

---

## Referee Report (RR1)

**Summary**

The manuscript presents a Hybrid Ship Clutter Identification (HSCI) algorithm that combines a random-forest classifier on six polarimetric/Doppler features with an adaptive sidelobe-masking scheme. While the authors have thoroughly resolved all minor issues and clarified methodological details, five core concerns remain only partially addressed, namely generalization to other waveforms/environments, annotation bias, quantitative precipitation-loss metrics, independent test datasets, and explicit code availability.

- **Rigorous Validation Metrics:** The authors have now added precision, recall, and F1-score metrics (all >97 %) in Section 4.2 and updated the abstract to include them, fully satisfying the request for rigorous performance reporting.

- **Generality Across Waveforms and Environments:** In Section 5 the authors acknowledge that this is a pilot study and explicitly discuss the limitation to LFM, 0.5° scans, but they have not provided any new empirical tests on NLFM waveforms, different elevation angles, or varied sea conditions. They must either include at least one additional validation case (e.g., an NLFM scan or a different elevation/season) or explicitly narrow all claims to the current test setup.

- **Manual Labeling Bias:** The authors clarified that a single annotator (the first author) labeled all gates with continuity checks, improving transparency in Section 2, but they still provide no inter-annotator agreement statistics. To address potential subjectivity, they should perform a simple second-annotator check on a representative subset or, at minimum, discuss in more depth how annotation bias might affect results.

- **Quantitative Assessment of Precipitation-Loss:** The manuscript now shows qualitative rain-field improvements in mixed scenes (Fig. 13b), but it lacks any numeric metric on how many precipitation gates are inadvertently removed. The authors need to report concise quantitative statistics (for example, the percentage of precipitation gates masked) for at least one mixed-scene case.

- **Independent Test Dataset:** The authors document using 5-fold GridSearchCV in Section 3.1.3, but they did not reserve any fully independent radar volume or day as a hold-out test. They should, if feasible, reserve one full sweep or day for independent validation and report its performance.

---

## Author Response (AR2)

**Response to Reviewer 1**

**Question 1:** Generality Across Waveforms and Environments

In Section 5 the authors acknowledge that this is a pilot study and explicitly discuss the limitation to LFM, 0.5° scans, but they have not provided any new empirical tests on NLFM waveforms, different elevation angles, or varied sea conditions. They must either include at least one additional validation case (e.g., an NLFM scan or a different elevation/season) or explicitly narrow all claims to the current test setup.

We agree with the reviewer's comment. To demonstrate the applicability of the HSCI algorithm in other scenarios, we have added a case using the NLFM waveform (Figures 11c and 11d) in the revised manuscript. This case was observed only one minute apart from the LFM waveform case shown in Figures 11a and 11b. A comparison of the reflectivity after ship clutter suppression reveals similar results, indicating that the HSCI algorithm maintains robust performance even under the NLFM waveform. The specific revisions can be found in Section 4.1 (Line 343 highlighted content).

[Figure]

**Figure 11. Reflectivity on 0.5° elevation of Kumpula radar using LFM (1310 UTC) and NLFM (1311 UTC) waveforms on 5 May 2020. (a) Before ship clutter filtering using LFM waveform; (b) After ship clutter filtering using LFM waveform; (c) Before ship clutter filtering using NLFM waveform; (d) After ship clutter filtering using NLFM waveform. The mainlobe identification results are highlighted by red circles.**

The occurrence of super-refraction which is conductive to ship clutter signatures has seasonal dependence (Lopez, 2009). Usually, ship clutter signatures are more common in summer than in winter. We have looked through our observations in the winter of 2020, but identified very limited ship clutter cases in precipitation. We found a single case where ship clutter was mixed with precipitation, with just one ship clutter present across the entire PPI. Fortunately, the HSCI algorithm successfully detected and filtered this ship clutter without excessively suppressing the surrounding precipitation echoes. Further clarification on this point has been added in DATA section of the manuscript (Line 126; highlighted content).

Lopez, P. (2009). A 5-yr 40-km-resolution global climatology of superrefraction for ground-based weather radars. Journal of applied meteorology and climatology, 48(1), 89-110.

[Figure]

**Figure xxx: 0.5° elevation of Kumpula radar using LFM waveform at 0010 UTC 6 Jau 2020. (a) Reflectivity before ship clutter filtering; (b) Reflectivity after ship clutter filtering; (c) Differential reflectivity before ship clutter filtering; (d) Differential reflectivity after ship clutter filtering; (e) Correlation coefficient before ship clutter filtering; (f) Correlation coefficient after ship clutter filtering.**

Although additional content has been included to help demonstrate that the applicability of the HSCI algorithm is not limited to a specific observational scenario, the authors still believe that the current study has not sufficiently validated the full extent of the algorithm's applicability. Therefore, the discussion of the algorithm's limitations has been further emphasized in Section 5 of the manuscript (Line 343 highlighted content). The specific content is as follows:

*Due to the limitations of available observational data, this study develops and evaluates the HSCI algorithm primarily based on ship clutter observed by the Kumpula radar in the Gulf of Finland, using an LFM waveform. While the features designed for ship clutter identification and the overall logic of the HSCI algorithm are intended to be general and not tied to a specific radar system or waveform, we acknowledge that the current validation is restricted to a single radar platform, geographic location, sea state, and waveform configuration. Therefore, the robustness and generalizability of the algorithm under varying conditions—such as across different radar systems, geographical regions, environmental conditions (e.g., sea clutter dynamics influenced by wind or temperature), seasons, and transmitted waveforms—remain to be comprehensively assessed. We recognize this as a significant limitation of the present work. Future studies will aim to address this issue by applying the HSCI algorithm to a broader range of observational datasets, enabling a more rigorous evaluation of its performance and adaptability in diverse operational scenarios.*

**Question 2:** Manual Labeling Bias

The authors clarified that a single annotator (the first author) labeled all gates with continuity checks, improving transparency in Section 2, but they still provide no inter-annotator agreement statistics. To address potential subjectivity, they should perform a simple second-annotator check on a representative subset or, at minimum, discuss in more depth how annotation bias might affect results.

We agree with the reviewer's suggestion. To further demonstrate the reliability of the dataset selection, the second author of this manuscript conducted a secondary annotation on 170 ship clutter cases from

10 scans, which were selected from the entire dataset (comprising 1,600 ship clutter cases across 110 scans). The results show that only two relatively weak ship clutter showed ambiguity, indicating that the overall annotation quality is high and that potential subjectivity has a limited impact on the dataset. The specific revisions can be found in Section 2 (Line 123 highlighted content).

**Question 3:** Quantitative Assessment of Precipitation-Loss

The manuscript now shows qualitative rain-field improvements in mixed scenes (Fig. 13b), but it lacks any numeric metric on how many precipitation gates are inadvertently removed. The authors need to report concise quantitative statistics (for example, the percentage of precipitation gates masked) for at least one mixed-scene case.

We agree with the reviewer's suggestion. In the revised manuscript, we replaced the original Figure 12 with a more informative case, in which several ship clutter events were completely mixed with precipitation echoes. This new case includes the three main output scenarios of the HSCI algorithm: (1) ship clutter is successfully identified (highlighted with red or white circles); (2) precipitation echoes are mistakenly identified as ship clutter (red or white circles indicated by arrows); and (3) ship clutter is not effectively identified (highlighted with black rectangles). The reviewer emphasized the second scenario. In this scenario, although errors occurred in the mainlobe identification, the application of velocity and SNR filters in sidelobe identification effectively controlled the loss of precipitation echoes—that is, only the range gate misidentified as a ship clutter mainlobe and its eight nearest neighboring gates were mistakenly removed. The specific revisions can be found in Section 4.1 (Line 343 highlighted content).

[Figure]

**Figure 12: 0.5° elevation of Kumpula radar using LFM waveform at 2240 UTC 4 June 2020. (a) Reflectivity before ship clutter filtering; (b) Reflectivity after ship clutter filtering; (c) Doppler velocity before ship clutter filtering; (d) co-polar correlation coefficient after ship clutter filtering. The mainlobe identification results are highlighted with red or white circles, with misidentified cases indicated by blue arrows, while the unrecognized ones are highlighted with black rectangles.**

**Question 4:** Independent Test Dataset

The authors document using 5-fold GridSearchCV in Section 3.1.3, but they did not reserve any fully independent radar volume or day as a hold-out test. They should, if feasible, reserve one full sweep or day for independent validation and report its performance.

We agree with the reviewer's suggestion. We used radar data from the same day (4 June) as that analyzed in Figure 13 for validation. We extracted 24 scans, selecting one scan per hour, and determined the locations of ship clutter using the same method as employed for dataset selection (described in Section 2). The evaluation results are summarized in Table 5. A total of 247 ship clutter events were identified across the 24 scans (i.e., the sum of the second and third columns in the second row). Of these, the HSCI

algorithm successfully detected 238, missing only 9. Additionally, 31 range gates associated with precipitation echoes were mistakenly identified as ship clutter. However, as discussed in the Question 3, the actual loss of precipitation echoes is well controlled, thanks to the velocity and SNR thresholds integrated into the HSCI algorithm. The specific revisions can be found in Section 4.2 (Line 426 highlighted content).

**Table 5: Performance evaluation results of HSCI algorithm for identifying ship clutter mainlobe based on observations from 4 June 2020.**

| Time (UTC) | Hit | Miss | False Alarm |
|---|---|---|---|
| Total | 238 | 9 | 31 |
| 00:00:41 | 21 | 0 | 2 |
| 01:00:40 | 14 | 0 | 2 |
| 02:00:40 | 10 | 1 | 4 |
| 03:00:41 | 11 | 1 | 7 |
| 04:00:40 | 14 | 0 | 2 |
| 05:00:40 | 12 | 1 | 0 |
| 06:00:41 | 8 | 0 | 1 |
| 07:00:40 | 9 | 0 | 2 |
| 08:00:40 | 7 | 1 | 0 |
| 09:00:41 | 4 | 0 | 0 |
| 10:00:41 | 2 | 0 | 1 |
| 11:00:41 | 6 | 1 | 1 |
| 12:00:41 | 10 | 1 | 0 |
| 13:00:41 | 16 | 0 | 3 |
| 14:00:40 | 14 | 0 | 2 |
| 15:00:40 | 16 | 0 | 0 |
| 16:00:40 | 11 | 0 | 0 |
| 17:00:41 | 8 | 1 | 0 |
| 18:00:41 | 12 | 0 | 2 |
| 19:00:40 | 10 | 0 | 1 |
| 20:00:41 | 11 | 0 | 0 |
| 21:00:41 | 4 | 0 | 0 |
| 22:00:41 | 6 | 1 | 1 |
| 23:00:40 | 2 | 1 | 0 |

---

## Author Response (AR3)

**Response to Reviewer 1**

**Question 1:** Line 126 (Page 6):

Clarify the phrase "only two relatively weak ship clutter showed ambiguity" slightly. Perhaps specify whether ambiguity relates to visibility, annotation consistency, or radar signal characteristics.

Agree. The ambiguity here primarily refers to the fact that the initial and secondary annotations could not reach a definitive agreement on the identification of these two relatively weak ship clutter cases.

We revised the original sentence "Specifically, 170 ship clutter from 10 scans were re-labeled, and only two relatively weak ship clutter showed ambiguity." to "Specifically, 170 ship clutter from 10 scans were re-labeled, and differences in labeling were observed only in two relatively weak cases between the initial and secondary annotations."

**Question 2:** Line 343 (Page 17):

"mainlobe identification employed the same random forest model as that used for the LFM waveform; however, the relative power relationship between the mainlobe and sidelobes was re-estimated based on observations under the NLFM waveform."

Suggest clarifying briefly why re-estimating sidelobe power relationships was necessary for the NLFM case. Was this driven by waveform differences that significantly impact sidelobe structure?

Agree. As correctly understood by the reviewer, the LFM and NLFM waveforms of the Kumpula radar exhibit different sidelobe structures (the relative power relationship between the mainlobe and sidelobes of the NLFM waveform is illustrated in the figure below). Accordingly, the revised manuscript includes the following clarification: "It is worth noting that the mainlobe identification employed the same random forest model as that used for the LFM waveform. However, the relative power relationship between the mainlobe and sidelobes was re-estimated based on observations under the NLFM waveform, primarily due to the different sidelobe structures of the two waveforms."

[Figure]

**Figure xxx: Statistical result of the relative power between the mainlobe and sidelobe for eight ship clutter events with high SNR.**

**Question 3:** Figure 11 (Page 17):

Recommend increasing the contrast or changing the color of the circles identifying the clutter mainlobes, as they are not easily distinguishable on some background echoes.

We attempted to replace the circle color with other options, but none proved to be more distinguishable against the light-colored background. Although black offers good contrast, it is already used in the figure to draw the range rings and radial lines for spatial reference. Therefore, we chose to thicken the red circles and submitted the high-resolution figure as an attachment, allowing readers to zoom in for better visibility while maintaining sufficient clarity.

[Figure]

**Figure xxx. Reflectivity on 0.5° elevation of Kumpula radar using LFM (1310 UTC) and NLFM (1311 UTC) waveforms on 5 May 2020. (a) Before ship clutter filtering using LFM waveform; (b) After ship clutter filtering using LFM waveform; (c) Before ship clutter filtering using NLFM waveform; (d) After ship clutter filtering using NLFM waveform. The mainlobe identification results are highlighted by red circles.**

**Question 4:** Line 370 (Page 18):

"although the presence of ship clutter can be inferred from polarimetric variables such as the ρhv (Fig. 12d), its impact on Zh (Fig. 12a) is so minimal that it is virtually undetectable."

Consider briefly discussing implications or recommendations for operational radar usage, given this minimal clutter scenario. Should these marginal cases be explicitly filtered operationally?

Agree. In the revised manuscript, we have added the following clarification: During the evaluation of the HSCI algorithm (Section 4.1), we observed that weak ship clutter affects overlapping precipitation echoes differently in reflectivity and polarimetric variables, with the latter being more susceptible to contamination. In such cases, a general-purpose non-meteorological echo filter based on a correlation coefficient threshold could be used as an optional supplement to the HSCI algorithm in operational settings, helping to mitigate the potential impact of residual weak ship clutter on meteorological applications.

**Question 5:** Table 5 (Page 22):

Suggest adding a brief note about the implications of the false alarm rate (31 gates mistakenly identified as ship clutter) for operational scenarios. Though limited, operational impacts or recommendations would enhance practical context.

Agree. In the revised manuscript, we have added the following clarification: In addition, the evaluation revealed that the HSCI algorithm has a tendency to over-identify ship clutter. However, since the algorithm employs velocity and SNR filters in the sidelobe identification, even when precipitation echoes are mistakenly identified as ship clutter, the resulting precipitation loss can be kept to a minimum. For these limited range gates that are incorrectly removed, interpolation techniques or speckle filters can effectively restore the missing information.

**Question 6:** Line 455 (Page 22):

"to move the stage of ship clutter identification and filtering from 'data processing' to 'signal processing'" – briefly clarify the expected improvement or practical benefits of this suggested future modification.

The primary advantage of moving ship clutter identification and filtering from the data processing stage to the signal processing stage lies in handling scenarios where ship clutter overlaps with precipitation. For example, in situations like that shown in Fig. 12, even if the ship clutter is accurately identified and properly removed, some precipitation echoes overlapping with the ship clutter may still be lost. In contrast, suppressing ship clutter at the signal processing stage allows the overlapping precipitation signals to be preserved.

In the revised manuscript, we have added the following clarification: In this study, the method used to suppress the negative impact of ship clutter is to mask radar variables at the range gates where ship clutter is identified. However, when precipitation echoes overlap with ship clutter (as shown in Fig. 12), this inevitably leads to the loss of precipitation data.

**Question 7:** Appendix (Page 24):

The estimation method for SNR based on reflectivity could benefit from a brief comment about the accuracy or reliability of this approach compared to directly measured SNR. Would inaccuracies here significantly affect the performance of the PSD identification?

The estimation of radar reflectivity typically involves two steps: first, estimating the signal power (or SNR—the difference lies in whether the noise power is normalized) from IQ data; and second, estimating the reflectivity factor from the signal power (or SNR). In terms of radar signal processing, the SNR is obtained prior to the reflectivity factor. Unfortunately, the Kumpula radar does not output the SNR as an intermediate product.

To address this, this paper proposes an SNR estimation method. As shown in the equations listed in the appendix, this method essentially reverses the second step of the reflectivity estimation process. Strictly speaking, the term "estimation" may not be entirely accurate—"derivation" would be more appropriate. Therefore, the precision of the derived SNR is the same as that of the SNR estimated from IQ data, without introducing any additional error.

In the revised manuscript, we have added further clarification on this issue: "It is worth emphasizing that the proposed SNR estimation method can be viewed as the inverse of the reflectivity estimation. As such, the derived SNR maintains the same level of accuracy as that obtained directly from the raw time series data, without introducing any additional errors".